# OCT4 enhances the firing efficiency of late DNA replication origins in mouse embryonic stem cells

Eddie Rodriguez-Carballo [1,2] ✉, Vasilis S. Dionellis[1], Sotirios G. Ntallis [1], Lilia Bernasconi[1], Ezgi G. Keskin[1,4] & Thanos D. Halazonetis [1,3] ✉

DNA replication initiates at specific genomic regions known as initiation zones (IZs), which follow a defined spatiotemporal program that is partially dependent on cell type. Here, we examine the replication-initiation patterns of pluripotent mouse embryonic stem cells (mESCs), which are characterized by a very short G1 phase and rapid entry into S phase. Using EdU-seq combined with cell-cycle synchronization and Repli-seq, we identify IZs that activate during S phase in mESCs and classify them as early, mid, or late according to the replication-timing (RT) domain to which they map. Remarkably, we find that some IZs mapping to mid or late RT domains activate within 1–2 hours of entry into S phase. Chromatin and nascent-transcriptome profiling reveal that these IZs associate with regions of open chromatin structure that are bound by the pluripotency factor OCT4. Transient OCT4 depletion reduces both chromatin accessibility and replication-initiation efficiency at these sites. These results provide an example of a pioneer factor, OCT4, facilitating DNA replication initiation by promoting local chromatin accessibility.

In mammals, genome duplication starts in a coordinated fashion at thousands of loci, the origins of replication. Origin firing is thought to take place in a controlled-stochastic manner, meaning that a given cell activates only certain origins among a defined available set[1]. Molecularly, any given origin will first be licensed, following a sequence of events that starts with binding of ORC onto chromatin and recruitment of a double-hexamer MCM helicase complex. In a next step, firing of licensed origins is stimulated by DDK and CDK protein kinases, which facilitate the assembly of firing factors and replisome subunits, including TRESLIN, MTBP, TOPBP1, MCM10, CDC45 and GINS[2]. The replication forks proceed bidirectionally, until they meet incoming forks, thus ensuring complete replication of the genome.

The genome is divided in replication timing (RT) domains, depending on when these domains are replicated during S phase. The RT program is established in G1 and shows variations between different cell types, although it shares some general features[3–6]. Early-replicating domains correspond to gene-rich regions, often actively transcribed, decorated with open chromatin marks and belonging to the so-called topological A compartment in Hi-C maps, whereas late-replicating domains correspond to poorly-transcribed, heterochromatic regions, belonging to the B compartment in Hi-C maps[7–9]. The relationship of the RT program to the functional regulation of the genome appears to be reciprocal, as early replication can influence the epigenome and act as a topological modulator and vice-versa[10–14]. One of the regulators at the interplay of replication timing and the epigenome is RIF1, which is commonly found in late replicating regions in association with the PP1 phosphatase, regionally counteracting DDK-driven origin firing[10,15].

Studying replication of the genome in higher eukaryotes is technically challenging. The SNS-seq and INI-seq methods map replication

[1]Department of Molecular and Cellular Biology, University of Geneva, Geneva, Switzerland. [2]Department of Preventive Dental Medicine and Primary Care. University Clinics of Dental Medicine (CUMD), University of Geneva, Geneva, Switzerland. [3]Department of Visceral Surgery and Medicine, Inselspital and Department for BioMedical Research, University of Bern, Bern, Switzerland. [4]Present address: IFOM ETS, the AIRC Institute of Molecular Oncology, Milan, Italy. ✉e-mail: edgardo.rodriguez@unige.ch; thanos.halazonetis@unige.ch

origins at very high resolution, but their results do not match well with those obtained by other methods[16,17]. Repli-seq and OK-seq have low resolution, meaning that they define broad (50–200 kb) replication initiation zones (IZs), where several origins might be found[18,19]. EdU-seq identifies origins at similar sites as Repli-seq and OK-seq and can also monitor fork progression. However, EdU-seq requires cell synchronisation and, to this date, only early-S initiation events have been studied by this method[20,21]. Unlike the origins identified in budding yeast, origins in higher eukaryotes do not appear to contain a consensus DNA sequence. Nevertheless, certain features, such as G4 structures and open chromatin marks, have been associated with origin activity[22–28].

Pluripotent stem cells constitute a powerful model to study DNA replication dynamics, because they are highly proliferative and have a very short G1 phase[29,30], which facilitates cell synchronisation and analysis of DNA replication. They also have a permissive chromatin environment that reflects their plasticity and pluripotency[31–33]. Their permissive chromatin state is maintained by pluripotency factors, among them OCT4, SOX2, NANOG and KLF4, which act as pioneer transcription factors[34–36]. Despite extensive work on replication origins and replication timing domains, the mechanisms that determine when and where origins are activated across the mammalian genome remain incompletely understood. In particular, the interplay between chromatin structure, transcriptional networks, and replication timing has not been fully resolved. Pluripotent stem cells provide a unique opportunity to address this gap, as their rapid cell cycles and transcription factor–driven chromatin state offer an ideal context to dissect how regulatory factors influence origin firing. In this study, we applied pharmacological and genetic approaches to show that pluripotency factors, and particularly OCT4, affect the replication timing program by facilitating firing of DNA replication origins located in late S phase replicating genomic domains. Thus, we propose that pluripotency factors affect not only the transcriptional program of mESCs, but also the efficiency of the late RT program, which is further regulated by CDC7, CDK1 and ATR.

## Results
### Replication timing landscape of mESCs

As a first approach to study the kinetics of genome duplication of mESCs we employed flow cytometry to determine the length of S phase. MEFs and bone-marrow mesenchymal stem cells (mMSCs) were also examined. For all three cell types, unsynchronized cells were pulsed with EdU for 30 min and their progression through the cell cycle was followed over time by flow cytometry. The EdU-positive cells reached G2 within 8–9 h in all three cell types (Supplementary Fig. 1a). Consistent with mESCs having very short G1 phases[30,37–40], the EdU-positive mESCs initiated a second round of DNA replication, ahead of the mMSCs and MEFs (Supplementary Fig. 1a; see vertical arrows at 9 and 12 h after the EdU pulse). In the same experiment, we also monitored progression through S phase of the EdU-negative cells. These cells entered S phase after the EdU pulse was administered and their progression through S phase, like that of the EdU-positive cells, could be monitored by the increase in their genomic DNA content (Supplementary Fig. 1a, see oblique arrows at 8 h). The profiles of the EdU-negative cells were also consistent with the length of S phase being similar in mESCs, mMSCs and MEFs.

Next, we employed high-throughput sequencing methods to monitor RT. First, we utilized a variant of Repli-seq, by labelling unsynchronized cells with EdU for 30 min and fractionating the cells based on genomic DNA content on a flow sorter (Supplementary Fig. 1b). Nascent DNA from these cells was conjugated to biotin, isolated and sequenced. From the Repli-seq profiles of pluripotent mESCs, MEFs and mMSCs, we determined the early, mid and late S phase RT domains at 10 kb resolution. These domains varied somewhat across the three cell types yet overlapped significantly (Supplementary Fig. 1c) and involved about similar fractions of the genome in all cell types (Supplementary Fig. 1d).

Repli-seq experiments extrapolate time from genomic content, meaning that the distribution of S-phase progression is based on the amount of the replicated genome but not on the time since S-phase entry. We, therefore, performed an EdU-seq time-course experiment, as a second approach to study RT. mESCs, synchronized in mitosis with nocodazole, were isolated by mitotic shake-off and then released in the cell cycle by washing nocodazole away. At various time points after exit from mitosis, EdU was added to the media and 30 min later the cells were harvested and processed for sequencing of their nascent DNA (Fig. 1a, b). MEFs and mMSCs could not be synchronized by mitotic shake-off, limiting the EdU-seq time-course analysis to the mESCs. As early as 2 h after mitotic exit, sharp peaks of EdU incorporation were evident, consistent with the earliest wave of origin firing and hence confirming the short G1 phase of mESCs. At this time point, we also observed a low but evident background signal specifically spreading along the late RT domains. Intriguingly, this signal could be attributed either to carryover late S phase cells or mitotic cells undergoing mitotic DNA synthesis or to actual replication being resumed in regions that were not properly replicated in the previous cell cycle, as suggested in a previous publication[41]. From 4 to 12 h after mitotic exit, the EdU incorporation profiles showed replication progressing from the early to the mid and the late S phase replication domains (Fig. 1c). Comparison of the Repli-seq (Supplementary Fig. 1c) and EdU-seq (Fig. 1c) profiles of mESCs revealed a high degree of similarity (Supplementary Fig. 1e).

### Late S phase IZs fire early in mESCs

The EdU-seq profiles of the mESCs revealed origin firing at multiple time points after mitotic exit (Fig. 1c). We expected that the IZs located within the mid and late RT domains would fire in mid and late S phase, respectively. However, at 4 h after mitotic exit, i.e. 2 h into S phase (corresponding to fraction S1 in the Repli-seq experiment; Supplementary Fig. 1e), we observed replication activity appearing, corresponding to putative IZs, within both the mid and late RT domains (Fig. 1c; green and orange arrowheads, respectively).

To determine whether the IZs that fired within the late RT domains in the above experiment were bona fide IZs, we examined mESCs synchronized under different conditions (Fig. 1d, e). First, we applied a CDC7 inhibitor (5 μM, named CDC7i and previously used by our group)[42] 3 h after mitotic exit, when the early origins had already fired, and examined nascent DNA synthesis by EdU-seq 4 h later. The CDC7i prevented the firing of the mid and late IZs, as should be the case for bona fide IZs (Fig. 1e, f: tracks (v) versus (x)). In a follow-up experiment, we released mESCs from the CDC7i block described above and monitored origin firing 2 h later. To obtain sharp EdU incorporation peaks and thus increase the resolution, we treated the cells with 2 mM hydroxyurea (HU) during the release period (Fig. 1e: track (xi)). Under these conditions, we observed the same IZs, mapping within mid and late RT domains, as the putative IZs identified previously in untreated cells 4 h after mitotic exit (Fig. 1c: track (ii)). The same IZs were also observed in cells synchronized by mitotic shake-off and treated with an inhibitor of the ATR checkpoint kinase to induce a burst in origin firing (Fig. 1e, f: track (xii)). From these results we concluded that the observed peaks in the mid and late RT domains were genuine IZs, albeit they had lower efficiency compared to IZs firing in early RT domains. We aimed to comprehensively map all IZs that were responsible for timed activation of replication along S-phase in mESCs. Using the datasets above and to avoid the inclusion of false positives, we called all bona-fide IZs under strict parameters (see methods and Supplementary Fig. 2a–c). We then classified them as early, mid or late (blue, green and yellow arrows, respectively, in Fig. 1f and Supplementary Fig. 2b) according to the RT domain (Supplementary Fig. 1d and top bar in Fig. 1c, f) to which they mapped (see methods and Supplementary Fig. 2a, b), showing very defined time dynamics (Supplementary Fig. 3a). In total, in chromosomes 1-X, we identified 5,162,

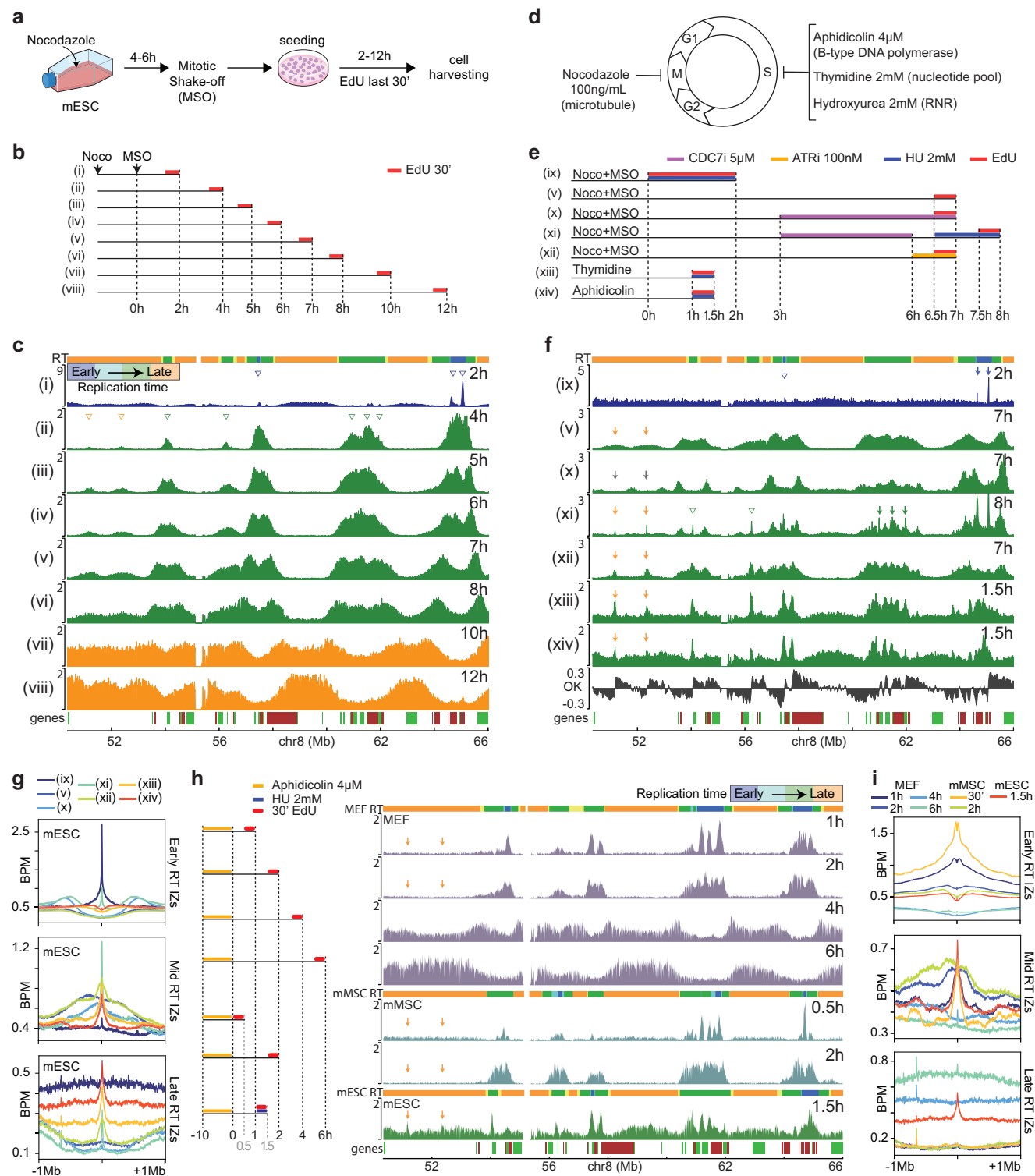

944 and 324 IZs mapping to early, mid and late RT domains, respectively (corresponding to bold arrows in Fig. 1). We next explored the dynamics of EdU incorporation associated with all the well-defined IZs. All of them contributed to the DNA replication profiles in accordance with the RT domain to which they belonged (Supplementary Fig. 2d).

Having established that well-defined IZs within late RT domains had fired in early-mid S phase in mESCs, we examined if we could obtain similar results with other cell synchronization methods. Thymidine and aphidicolin were used to induce cell cycle arrest mostly at the G1/S boundary. Within 1.5 h of release from a thymidine or

aphidicolin-induced arrest, we observed origin firing within the late RT domains (Fig. 1e, f: tracks (xiii, xiv) and Supplementary Fig. 3b). Importantly, the IZs identified by all the experiments described above align well across experiments and also with the IZs identified in mESCs by the OK-seq method (Fig. 1f; compare track OK[18] to tracks (ix-xiv) and Supplementary Fig. 3c).

Next, we investigated if the firing of late RT IZs in early-mid S phase was a specific feature of mESCs. We were unable to synchronize MEFs and mMSCs by mitotic shake-off; however, we could synchronize these cells with aphidicolin (Supplementary Fig. 4a). Neither MEFs, nor

**Fig. 1 | Identification of early, mid and late initiation zones (IZs). a** Mitotic shake-off (MSO) scheme. **b** Experimental outline of the EdU-seq time-course experiment (tracks i-viii, 2–12 h after MSO). EdU was added 30 min before harvesting the cells. **c** EdU-seq profiles at the indicated time points after mitotic exit. The EdU-incorporation profiles are colored blue, green or orange depending on whether the cells were in early, mid or late S phase, respectively, at the indicated time point after mitotic exit. Select putative IZs mapping to early, mid and late replicating regions are indicated by blue, green and orange open arrows, respectively. Top: RT (replication time) domains, as determined by Repli-seq (early, mid and late domains are colored blue, green and orange, respectively). Bottom: gene annotation (forward and reverse direction of transcription are marked by green and red boxes, respectively). **d** Scheme showing the agents used to synchronize mESCs in mitosis or at the G1/S boundary. **e** Experimental outline used to identify IZs at mid and late RT domains. ATRi (ATR inhibitor BAY-1895344 100 nM), HU (hydroxyurea 2 mM), CDC7i (CDC7 inhibitor 5 μM). **f** EdU-seq profiles in cells released into S phase from the G1/S boundary. The EdU-seq profile in track (v) is reproduced from panels (**b**) and (**c**). Tracks (ix-xiv) correspond to the experiments outlined in panel (**e**). Track OK shows origin-mapping profiles identified by the OK-seq method[18]. Select bona-fide IZs are indicated by vertical blue, green and orange arrows, respectively. Open arrows show IZs that were not considered by the stringent cut-off. RT domains and gene annotation are as in panel (**c**). **g** Average (BPM) EdU-seq profiles around the IZs called in early ($n = 5304$), mid ($n = 985$) and late ($n = 347$) RT domains for each of the experiments shown in panel (**f**). **h** EdU-seq profiles of MEFs, mMSCs and mESCs synchronized with aphidicolin, released and collected at different time points. Orange arrows show the positions of IZs in late replicating domains present only in mESCs (also shown in panels **c** and **f**). RT domains for each cell type are shown above their respective EdU-seq profiles. **i** Average (BPM) EdU-seq profiles of MEFs, mMSCs and mESCs from the experiments shown in panel (**h**), centered around the mESC IZs called in early ($n = 5162$), mid ($n = 944$) and late ($n = 324$) RT domains.

mMSCs fired origins within late RT domains 2 h after release from an aphidicolin block (Fig. 1g, orange arrows and Supplementary Fig. 4b). Instead, 4–6 h after release from the aphidicolin block, MEFs exhibited broad regions of EdU incorporation in the late RT domains, suggesting different patterns of origin firing from cell to cell in these domains. Comparable results could be observed in similar late RT domains when looking at the Repli-seq data (Supplementary Fig. 1c). Thus, firing of well-defined late RT IZs in early-mid S phase was a specific feature of mESCs.

### The RT program of mESCs is not affected by nucleotide levels

Increased levels of deoxynucleotide triphosphates (dNTPs) stimulate premature firing of late origins in yeast[43], prompting us to examine if firing of late RT IZs in early-mid S phase in mESCs was linked to dNTP levels. First, we supplemented the tissue culture media with nucleosides (Embryomax). This intervention did not change the RT program, as determined by both Repli-seq and EdU-seq (Supplementary Fig. 5).

In a reciprocal experiment, mESCs were treated with low doses of HU (100 μM) to impede dNTP production. The HU treatment slowed progression through S phase (Fig. 2a and Supplementary Fig. 6a–c) and lowered fork speed rates (Fig. 2b, c). However, the replication profiles, determined at various times after mitotic exit, showed no change in the RT program, other than the overall delay in S phase progression. As an example, the EdU-seq profile 6 h after mitotic exit in the untreated cells was identical to the 10 h profile of the HU-treated mESCs (Fig. 2a and Supplementary Fig. 6d–g).

### Interplay between ATR, CDC7 and CDK1 in regulating late origin firing

The DNA replication checkpoint has been reported to control the timing of late origin firing via pathways that involve CDC7, RIF1 and CDK1. The main effector of the DNA replication checkpoint is the kinase ATR[44]. Upon entry into S phase, exhaustion of dNTPs, as a result of DNA synthesis, activates ATR, which triggers dNTP synthesis[43,45,46]. In parallel, ATR suppresses firing of more origins until appropriate dNTP levels are reached. The molecular mechanism involves phosphorylation of DBF4, the regulatory subunit of CDC7[47,48], and activation of RIF1, which associates with Protein Phosphatase 1 at late origins to antagonize CDC7 activity[49–52].

We wondered whether mESCs lacked a functional DNA replication checkpoint, as this could explain the firing of late IZs in early-mid S phase. As a first step to answer this question, we determined whether the DNA replication checkpoint is activated in unperturbed mESCs during S phase by monitoring phosphorylation of histone H2AX and the kinase CHK1, both of which are ATR substrates[44,53]. Consistent with previous reports[41], at 6 and 10 h post-mitotic exit, we observed increased levels of phosphorylated H2AX (γH2AX) and CHK1 (p-CHK1) in both untreated mESCs and mESCs treated with low doses of HU (Fig. 2d). In subsequent experiments, inhibition of ATR by a specific chemical inhibitor (ATRi; BAY-1895344) stimulated firing of mid and late RT IZs, both in the presence and absence of low doses of HU (Fig. 2e–g; green and orange arrows for mid and late RT IZs, respectively). We conclude that the DNA replication checkpoint is not defective in mESCs and that it curtails firing of mid and late RT IZs, making ATR a putative rheostat for the transition into the late replication program.

In addition to the DNA replication checkpoint, firing of late origins is regulated by the CDC7-RIF1 axis and by CDK1[49,50,52]. To study the interplay between ATR, CDC7 and CDK1, we synchronized mESCs by mitotic shake-off and 3 h later treated them with TAK-931 (1 μM), a highly specific CDC7 inhibitor, different from our previously used CDC7i, and which has been used in clinical trials. EdU-seq profiles from cells harvested 4 and 7 h after mitotic exit revealed that, similarly to our CDC7i, TAK-931 suppressed firing of origins at the mid and late RT domains (Fig. 3a, b). In agreement with the DNA replication checkpoint antagonizing CDC7, inhibition of ATR rescued firing of these origins in mESCs treated with TAK-931 (Fig. 3a, b). Moreover, the rescue of origin firing was suppressed by a CDK1 inhibitor[54] (Fig. 3a, b), consistent with reports that ATR inhibitors activate CDK1, which in turn induces origin firing by inhibiting RIF1 and by phosphorylating the MCM helicase (Fig. 3c)[49–52,55–58].

### A high fraction of the genome is transcriptionally active in mESCs

Most transcriptionally active genomic domains are replicated in early S phase[2,3,59–61]. Therefore, the firing of late IZs in early-mid S phase in mESCs might be related to a higher fraction of the genome being transcriptionally active in mESCs, as compared to differentiated cells or to specific global transcriptional dynamics along the cell cycle. To address these possibilities, we monitored nascent transcription in mESCs and MEFs. The uridine analogue 5-ethynyl-uridine (EU) was added to cultures of asynchronous cells for 30 min; the cells were then sorted by their genomic content into six fractions, as done for Repli-seq, and nascent transcription was determined genome-wide for each fraction by EU-seq or by FACs (Fig. 4a and Supplementary Fig. 7a, b).

The genome-wide averages of EU signals around all early, mid and late RT IZs in mESCs revealed high levels of nascent transcription flanking the early RT IZs, a modest signal around the mid RT IZs and practically no signal around the late RT IZs (Fig. 4b). We note that there was a sharp drop of nascent transcription signal at the very center of the early and mid IZs (Fig. 4b), which can be explained by replication origins mapping to transcriptionally silent intergenic regions next to actively transcribed genes[9,62–65].

We compared the nascent transcription between mESCs and MEFs, revealing that the global output was indistinguishable along the cell cycle when analyzing mESCs and MEFs by FACs (Supplementary Fig. 7a, b). Nevertheless, a detailed analysis of transcriptional dynamics by EU-seq experiments showed clear differences. For each cell type, we

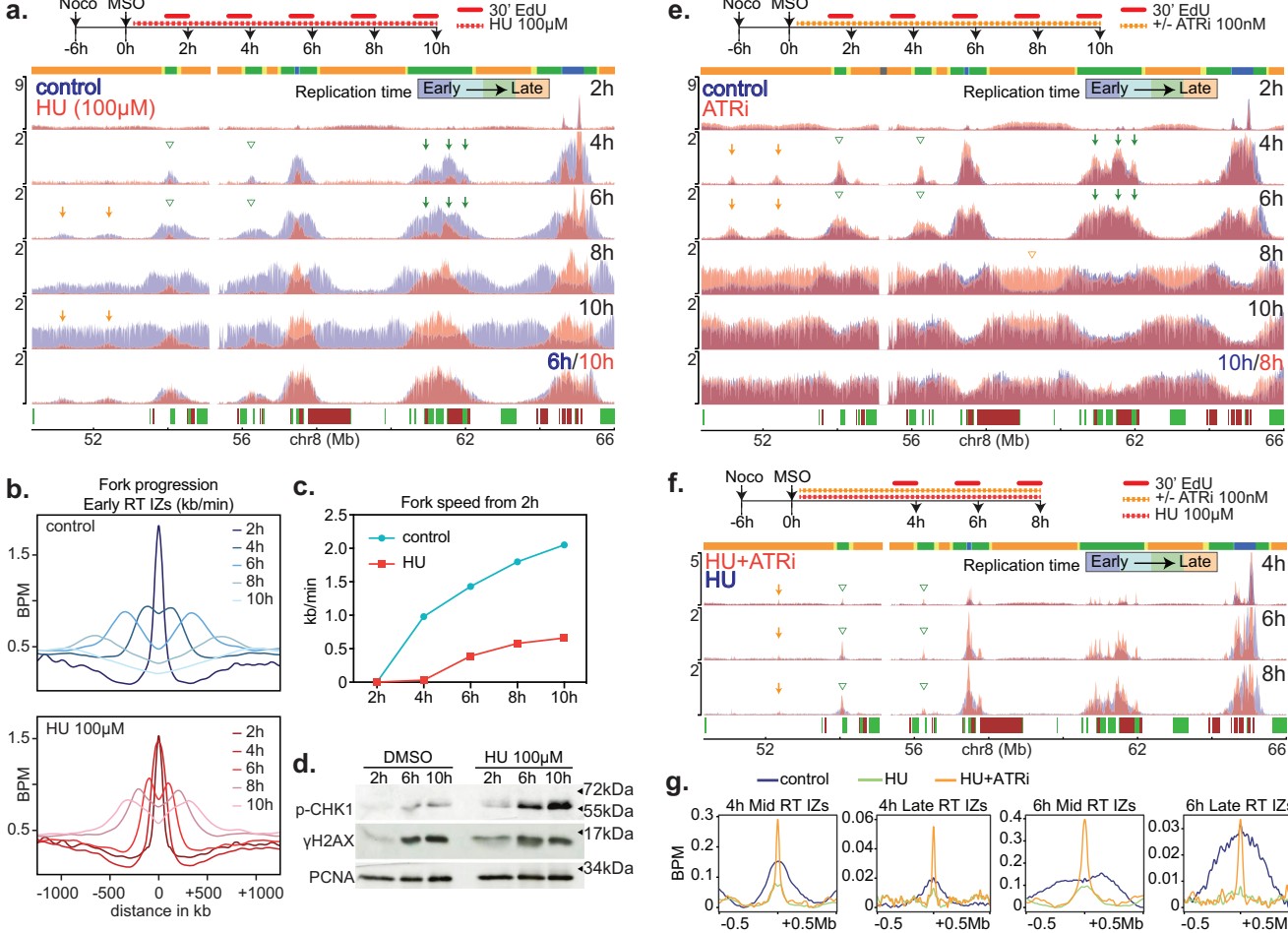

**Fig. 2 | Hydroxyurea and ATR effects on firing of late origins in mESCs. a** Overlay of EdU-seq plots of control (blue) and HU-treated (red, 100 μM hydroxyurea) mESCs at different time points after mitotic shake-off (MSO). Note that the late RT IZs present at 6 h in control cells (orange arrows) could only be spotted at 8 and 10 h in the HU profiles. Above, experimental design of the experiment; EdU was added for the last 30 min before harvesting the cells. RT domains and annotated genes are as in Fig. 1c. **b** Average EdU-seq profiles around a set of isolated early origins (n = 380) in control (top) and HU-treated (bottom) cells at the indicated time points after MSO. **c** Fork speeds calculated from the average EdU-seq profiles shown in (**b**). The profile at 2 h served as the starting point from which fork speeds were calculated for the later time points. **d** Western-blot of mESCs at 2, 6 and 10 h after MSO. The cells were treated with DMSO or HU. Molecular weight markers are shown on the right of the blot. The experiment was performed once. Source data

are provided as a Source Data file. **e** Overlay of EdU-seq profiles of control (blue) and ATRi-treated (red, 100 nM ATR inhibitor) mESCs. Green and orange arrows at 4 h and 6 h denote well-defined mid and late RT IZs. The orange open arrow at 8 h shows earlier replication of a late domain in ATRi-treated cells. RT domains and gene annotation are indicated as in Fig. 1c. The experimental scheme is shown above the EdU-seq profiles. **f** Overlay of EdU-seq profiles of mESCs treated with hydroxyurea (HU, blue) or HU plus ATRi (red) at 4, 6 and 8 h after MSO. Mid and late RT IZs that fired only in the HU+ATRi-treated cells are indicated by green and orange arrows, respectively. RT domains and gene annotation are as in Fig. 1c. The experimental scheme is shown above the EdU-seq profiles. **g** Average (BPM) EdU-seq profiles around mid and late RT IZs at 4 and 6 h after MSO in control, HU and HU plus ATRi-treated mESCs.

clustered all the annotated transcripts according to nascent transcription levels, flow sorting fraction (based on genomic DNA content) and RT (Fig. 4c). Regarding the distribution of transcriptional units along replication domains, mESCs had more genes located in the mid and late RT domains compared to MEF (Supplementary Fig. 7c). Generally, in both mESCs and MEFs, early RT domains were enriched with mid and highly transcribed genes, whereas genes located in late replicating domains showed low to negligible levels of expression (Fig. 4c, d and Supplementary Fig. 7c). Specific quantification of the differences in nascent RNA products showed a general upregulation of all transcripts in mESCs compared to MEFs. These differences were statistically more significant in earlier replicated gene units (Supplementary Fig. 7c). Thus, transcriptional activity could partially explain why firing of late S IZs differs between mESCs and MEFs, although transcription alone does not seem to be the only determinant of replication time[66].

## Epigenetic marks at IZs

We investigated whether efficient initiation of replication was solely linked to transcription or if other genomic features could also account for it. Distance correlation between our mESCs IZs datasets and annotated TSS and TTS, which have been previously associated to origins of replications[9,20,67], showed increasing values from early to late replicating domains (Fig. 4e). Nevertheless, when focusing on a compilation of known mESCs enhancers[68], the associated distance was nevertheless substantially reduced for late RT IZs (Fig. 4e). We further examined whether IZs were associated with features of open chromatin. To address this possibility we analyzed publicly available mESCs epigenetic data[69–74]. We saw that early, mid and late RT IZs were distributed differently across lamina-associating domains (LADs)[8,73]. Thus, most early and mid IZs mapped to broad inter-LAD regions; whereas, the late RT IZs were mostly located within LADs (Fig. 4f).

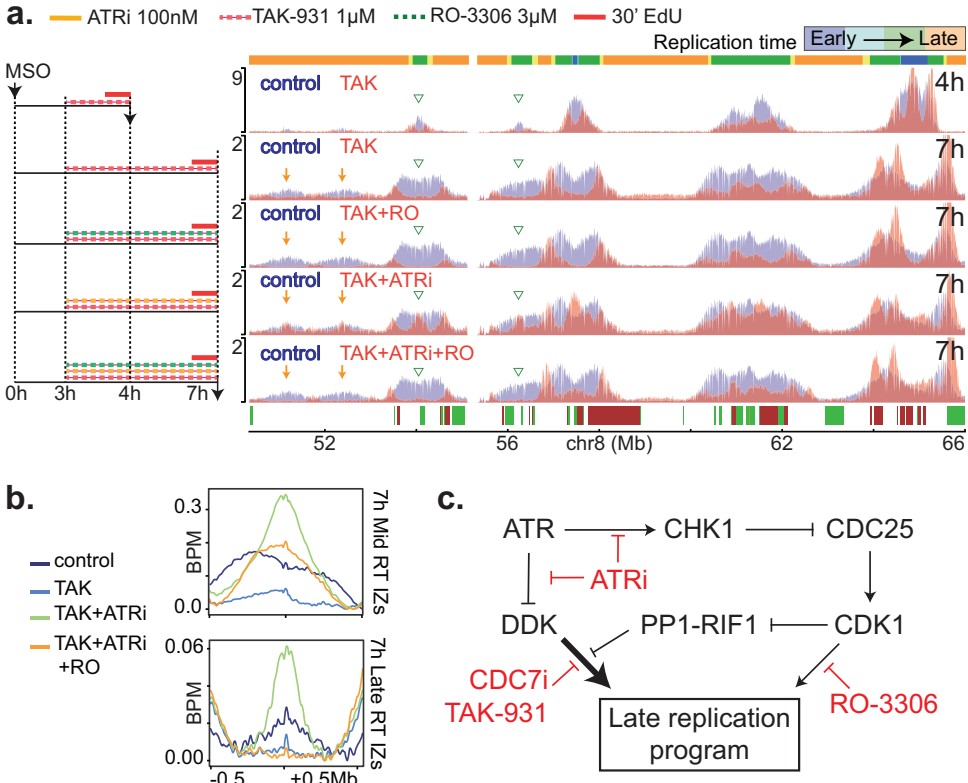

**Fig. 3 | DDK and CDK1 induce firing of late RT IZs. a** EdU-seq profiles of mESCs treated with the CDC7 inhibitor TAK-931 (1 μM, TAK), ATR inhibitor (ATRi 100 nM) and/or CDK1 inhibitor (RO-3306 3 μM, RO). Left, experimental outlines; right, corresponding EdU-seq profiles. Orange arrows and green arrowheads correspond to late and putative mid RT IZs, respectively, that are specially affected. RT domains and gene annotation are indicated as in Fig. 1c. **b** Average EdU-seq profiles around mid and late RT IZs 7 h after MSO for control mESCs and mESCs treated with the indicated inhibitors. **c** Molecular pathways by which ATR regulates late origin firing. DDK CDC7 and DBF4 complex, PP1 protein phosphatase 1.

The early IZs were decorated with active histone variants (H3.3 and H2AZ) and euchromatic marks (H3K4me1 and H3K27ac) (Fig. 4g and Supplementary Fig. 8). H4K20me2, which is recognized by the BAH domain of ORC1, was also enriched at the early IZs (Fig. 4g), as were G4 quadruplexes and open chromatin regions identified by the Assay for Transposase-Accessible Chromatin (ATAC-seq) (Fig. 4h, i). The same marks that were enriched at the early IZs were also enriched at the mid and late RT IZs, although the signal intensity was lower (Fig. 4g, h, i and Supplementary Fig. 8). Thus, open chromatin features are present at IZs and their signal intensity correlates with RT.

**The pluripotency factor OCT4 acts as pioneer factor for mid and late RT IZs**

Given the presence of open chromatin marks at IZs, we wondered whether the firing of late RT IZs in early-mid S phase might be driven by recruitment of mESC-specific pioneering factors. Likely candidates would be the pluripotency factors OCT4, SOX2, KLF4 and NANOG, which are expressed in mESCs, but not in differentiated cells. Implicit in this hypothesis is that the abovementioned proteins bind to the genomic sites, where the mid and late RT IZs are located. Analysis of publicly available datasets[69] revealed that these pluripotency factors are enriched at early, mid and late RT IZs (Fig. 4j, k and Supplementary Fig. 8).

Out of the above pluripotency factors, we focused on OCT4 by studying the well-characterized ZHBTc4 stem cells, in which OCT4 levels are regulated by a tet-off system (Fig. 5a)[75]. Treatment of these cells with doxycycline for 12 or 18 h suppressed OCT4 levels (Supplementary Fig. 9a) without affecting the kinetics of entry into S phase after synchronization by MSO (Supplementary Fig. 9b). To monitor firing of mid and late RT IZs, the ZHBTc4 cells were treated with or without doxycycline and synchronized by MSO or with thymidine. Samples were collected at several time points after release from the cell cycle block and processed for EdU-seq (Fig. 5b). Inspection of representative genomic regions revealed delayed firing of mid and late RT IZs in the absence of OCT4 (Fig. 5c, d and Supplementary Fig. 9c). The affected IZs corresponded to sites where OCT4 bound to regulate chromatin accessibility, as documented from publicly available datasets[69] (Fig. 5c, d and Supplementary Fig. 9c).

We aimed to study the genome-wide effect of OCT4 on the initiation of replication analyzing EdU-seq signal 4 h post-MSO at all early, mid and late RT IZs in ZHBTc4 cells treated with or without doxycycline (OCT4-OFF and OCT4-ON, respectively). For this purpose, we performed linear regression analyses comparing, for each group of early, mid and late initiation zones, the change in EdU signal levels using log-transformed OCT4-ON signal as the independent variable. This quantification revealed a significant positive correlation between EdU signal and the presence of OCT4, with the slope become markedly steeper in late replicating IZs (Fig. 5e). The duration of the treatment with doxycycline, 18 or 12 h, did not affect the results (Fig. 5e). A different quantification, plotting the OCT4-ON and OCT4-OFF log2 signal, provided similar results, showing a higher deviation from the expected values in LS-IZs (Supplementary Fig. 9d, e).

The same regression analysis and scatter plots of the ATAC-seq signal from public datasets paralleled the EdU-seq data, as they revealed decreased chromatin accessibility at late RT IZs following OCT4 depletion, which was more pronounced around late RT IZs (Fig. 5f and Supplementary Fig. 9f). Considering that OCT4 was present at most of the genomic sites corresponding to mid and late RT IZs (as examples see the sites labelled IZ34 and IZ128 in Fig. 5d and Supplementary Fig. 9d, e), our results suggest that OCT4 enhances chromatin

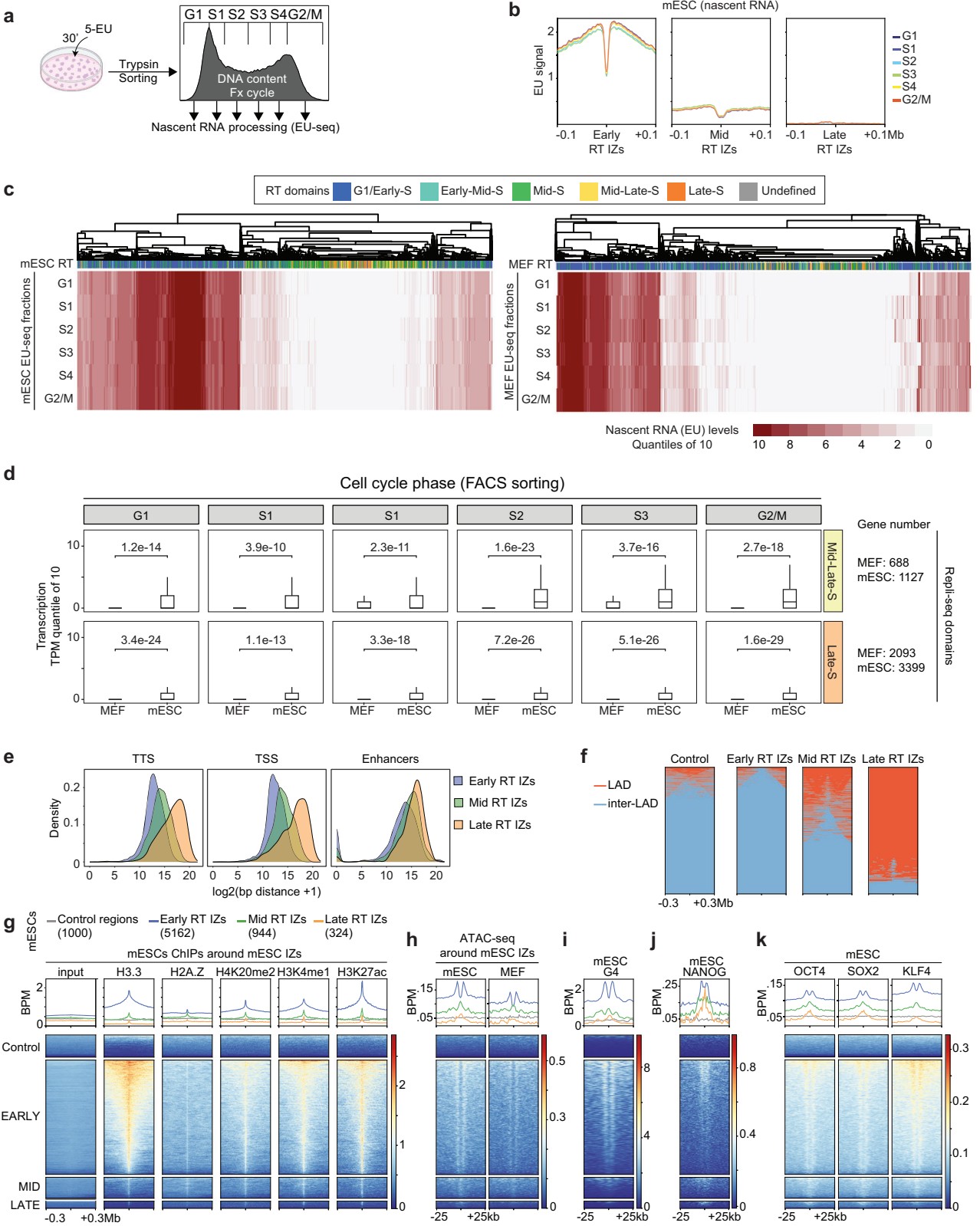

accessibility at mid and late S phase IZs and in doing so increases origin firing efficiency, thus promoting their activation in early S phase.

## Discussion

In this study, we investigated the replication dynamics of pluripotent cells. Remarkably, we observed that OCT4 stimulates the firing of origins within late replicating genomic domains, suggesting that

pluripotency and/or pioneer factors may have functions that extend beyond regulating transcriptional activity.

mESCs provide a unique system to study DNA replication IZs. Because the G1 phase of these cells is only about 2 h long, mESCs enter S phase rather synchronously after mitotic exit. This allowed us to use the EdU-seq method[21] to monitor origin firing upon entry of the cells into S phase and at later time points, without having to treat the cells

**Fig. 4 | Transcription patterns and open chromatin marks at RT IZs.**
**a** Experimental outline of EU-seq experiments with sorted cells. EU was added for 30 min to non-synchronized cells (mESCs or MEFs); the cells were then sorted in six fractions according to DNA content (compare to Supplementary Fig. 1b). **b** Average EU-seq signal around early, mid and late RT IZs for each fraction of sorted mESCs. **c** Gene clustering according to nascent transcription along the cell cycle in mESCs (left) and MEFs (right). The genes (displayed in vertical columns) were clustered according to their relative expression levels in the six cell cycle fractions (rows G1 to G2/M) and RT (first row of the heatmap). **d** Transcriptional quantile of genes located in mid-late S (MSLS, top) and late S (LS, bottom) RT domains for MEFs and mESCs. Values represent the FDR corrected p-values of unpaired t-test of one experiment comparing mESC and MEF. Box plots show the median (centre line), the 25th (Q1) and 75th (Q3) percentiles (bounds of the box), and the minimum and maximum values within 1.5× the interquartile range (whiskers). Points outside this range are plotted individually as outliers. Source data are provided as a Source Data file. **e** Distribution of the closest transcription start site (TSS), transcription termination site (TTS) and mESC enhancers(enhancer ATLAS v2.0[68]) to each early, mid and late mESC RT IZs. Source data are provided as a Source Data file. **f** Heatmaps showing lamina-associating domains (LADs) and inter-LAD regions[73] around early, mid and late RT IZs. **g** Heatmaps showing ChIP-seq signal intensities for the indicated histone marks around all early, mid and late RT IZs from published mESC datasets. The number of plotted early, mid and late RT IZs are shown in parentheses. **h–k** Heatmaps of ATAC-seq signals from mESC and MEF (**f**), G4 quadruplexes (**e**) and ChIP-seq signals of NANOG (**h**), OCT4, SOX2 and KLF4 (**i**) from published mESC datasets at early, mid and late RT IZs.

with HU. As a result, we observed efficient origin firing within the mid and late RT genomic domains and were able to map these IZs on the genome with high resolution.

Our results shed light on general principles defining the initiation of DNA replication and the establishment of the RT program (Fig. 6). In mESCs, DNA replication was mediated by well-defined IZs. The early RT IZs were very efficient and fired shortly after entry into S phase. The firing of mid RT IZs was evident at 3 h and continued up to 8 h after mitotic exit, indicating a lower efficiency of origin firing that took several hours to complete. The late RT IZs were even less efficient with firing evident in few cells as early as 4 h and continuing beyond 8 h after mitotic exit. We understand that the bulk approach of our study, combined with the fact that mESC cultures may host different cell populations, limits the capacity of our results to fully dissect the relationship between replication time scheduling and replication firing efficiency. Nevertheless, our results fit into the "controlled-stochastic" model of eukaryotic DNA replication, a model supported by single-cell and single-molecule approaches, which posits that the efficiency of origin firing is a determinant of RT[1,76–82].

During our IZ calling, to avoid false positives, we applied strict calling parameters on the EdU-seq signal, using different signal thresholds for each class of IZ and an exclusion zone extrapolated from fork speed, to prevent confusion from signals originating from neighboring ongoing forks. This approach limits the proportion of false positives, yielding bona fide lists of IZs associated with specific RT domains. Unavoidably, this strategy restricts our high-resolution IZ identification to replication activity coming from very efficient origins. For the same reasons, we note that a substantial number of efficient IZ false negatives remain unexplored in our dataset. These false negatives are expected to occur predominantly in the transition zones of RT domains - regions that are inherently more heterogeneous in their replication timing and often overlap with our exclusion zones, making their detection under stringent criteria more challenging.

Hence, we acknowledge that we did not fully elucidate the mechanisms driving the higher firing efficiency of early RT IZs compared to the less efficient mid and late RT IZs. Nonetheless, our study provides useful insights into this phenomenon, laying the groundwork for future mechanistic investigations. We show that firing efficiency correlated with transcription and chromatin accessibility, as the signal intensity of open chromatin marks correlated well with RT and, by inference, with firing efficiency. In yeast, it is documented that an open chromatin structure contributes to origin firing efficiency by increasing the probability of MCM recruitment[83–85].

Interestingly, not all late replicating genomic domains in mESCs were associated with a late RT IZ, as some late RT genomic domains were replicated 10–12 h after mitotic exit without well-defined IZs being evident at earlier time points. An example is the late RT domain located at the center of the representative genomic region shown throughout this study (chr8, Mb 58–60; Fig. 1c, f), where replication appears to initiate simultaneously across this 2 Mb region (Fig. 1c, f). This pattern suggests the presence of multiple low-efficiency origins throughout late domains (Fig. 6). Under one scenario all these origins could fire simultaneously in every cell or, more likely, a few of these origins would fire in each cell in a stochastic manner (Fig. 6). Since EdU-seq monitors origin firing in a population of cells, we cannot discriminate between these possibilities.

The existence of two ways by which late RT domains are replicated in mESCs is supported by data obtained by OK-seq and single-cell methods[77,80–82,86]. The late RT domains with well-defined IZs by EdU-seq match OK-seq data showing origins at the same genomic positions (for example, Fig. 1f; chr 8, Mb 51–53), whereas no origins were evident by OK-seq in the late RT domains that lacked well-defined IZs by EdU-seq (for example, Fig. 1f; chr 8, Mb 58–60).

Unlike mESCs, the late RT domains in MEFs appeared to be replicated exclusively by low-efficiency stochastic origins (Fig. 1h, i). We attribute the presence of well-defined late RT IZs in mESCs to pioneer factors, such as OCT4, that enable specific genomic sites to serve as IZs, presumably by locally opening the chromatin structure. We cannot completely rule out unknown indirect effects of OCT4 on cell-cycle progression. OCT4 has been shown to transcriptionally regulate p21[87] and to prevent premature mitotic entry by inhibiting CDK1 activation during G2[88]. However, in our study, we depleted OCT4 acutely so as to confine any effects to a single cell cycle as much as possible, and we did not observe a marked change in S-phase entry. Therefore, the hypothesis we propose is consistent with previous studies showing that transcription factors can stimulate origin activity in diverse organisms ranging from viruses to yeast and metazoans[89–97]. Moreover, the acute depletion of the pluripotency factor OCT4 in mESCs led to decreased chromatin accessibility[34,35], concurrent to decreased origin firing efficiency at the well-defined late RT IZs (Fig. 5).

Beyond the local determinants, origin firing in mESCs was regulated by kinases that regulate cell cycle progression, including CDC7 and CDK1, and kinases that signal the presence of DNA replication stress, notably ATR. Inhibition of ATR increased origin efficiency, even when CDC7 was inhibited; under these conditions, origin firing was dependent on CDK1 activity. Our results are, thus, consistent with molecular pathways already described in yeast and mammalian cells that link CDC7, CDK1 and ATR activities[41,44,47,49–52,55–58,98–101].

In conclusion, our work suggests that RT is determined by the firing efficiency of well-defined origins and by global regulators, including the DNA replication checkpoint and cell cycle-regulated kinases. In mESCs, we have documented the presence of well-defined IZs, even in late replicating genomic domains. The presence of these efficient IZs is dependent on expression of the pluripotency factor OCT4, which act as a pioneering factor to open chromatin structure locally. The functional significance of these well-defined IZs in late RT domains in mESCs remains to be determined.

## Methods
### Cell culture
Mouse embryonic stem cells (mESCs) from 129/SVEV strain were purchased from Merck (cat. no. CMTi-1, passage 11). They were thawed,

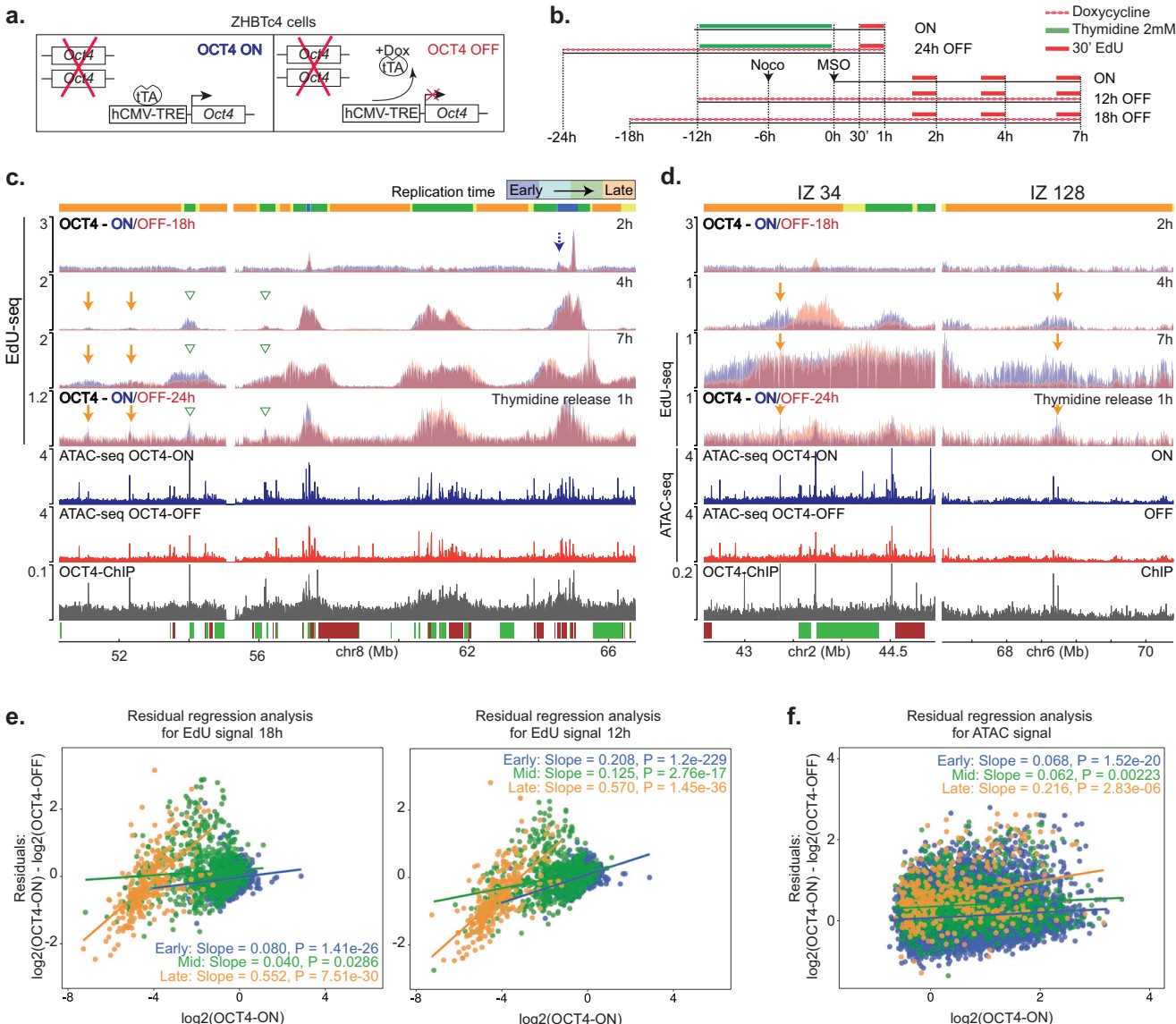

**Fig. 5 | OCT4 enhances firing of select mid and late RT IZs in mESCs. a** Scheme of the tet-off ZHBTc4 cells, in which OCT4 levels can be depleted by adding doxycycline (Dox). **b** Outline of EdU-seq experiments performed with ZHBTc4 cells synchronized with thymidine or by mitotic shake-off (MSO). To suppress OCT4 expression, treatment of the cells with doxycycline was initiated 24 h before release from the thymidine block or 12 or 18 h before the MSO and kept until the cells were harvested. **c** EdU-seq profiles of ZHBTc4 cells at the indicated times after MSO or after release from a thymidine block. Profiles with OCT4 expression turned ON or OFF (doxycycline added 18 h before MSO) are compared. Plots below the EdU-seq profiles: ATAC-seq profiles from asynchronous ZHBTc4 cells with OCT4 expression turned ON or OFF[34] and OCT4 ChIP-seq profile from asynchronous mESCs[105]. Select RT IZs mapping to mid and late RT domains are indicated by green and orange arrows, respectively. RT domains and gene annotation are indicated as in Fig. 1c.

**d** EdU-seq, ATAC-seq and OCT4 ChIP-seq profiles for the genomic regions spanning two selected IZs (34 and 128), shown as in panel (**c**). **e** Linear regression analysis of EdU signal at early (blue), mid (green) and late (orange) RT IZs of ZHBTc4 cells after depletion of OCT4 for 18 h (left) or 12 h (right). The EdU-seq signal for each IZ was computed as BPM in a region +/−5 kb around the centre of the IZ and residuals from log2 values were computed between cells expressing (ON) versus not expressing (OFF) OCT4. Log-transformed OCT4-ON signal was used as the independent variable. **f** Linear regression analysis of ATAC signal around early (blue), mid (green) and late (orange) RT IZs of mESC. The ATAC signal for each IZ was computed as BPM in a region +/−5 kb around the centre of the IZ and residuals from log2 values were computed between cells expressing (ON) versus not expressing (OFF) OCT4. Log-transformed OCT4-On signal was used as the independent variable. The ATAC-seq data are from ref. 34.

amplified as per manufacturer instructions for three passages, aliquoted and frozen. This stock of cells was routinely used on gelatin plates (or flasks) for no more than 32 passages. For all experiments, mESCs were grown in DMEM+Glutamax (Gibco, 61965-026), 10% ES-FBS (Gibco 16141-079), MEM NEAA (Gibco 11140-035), sodium pyruvate (Gibco, 11360-039), 1 mM beta-mercaptoethanol (Gibco, 31350-010), penicillin/streptomycin (P/S, Gibco, 15140-122), 100 ng/mL LIF (stocked at 1-2 mg/mL; produced at the Protein Production and Purification Platform of the EPFL, Lausanne, Switzerland), 3 μM GSK3 inhibitor (Sigma 361559) and MEK inhibitor (PD0325901,

Selleckchem). ZHBTc4[75] are tet-off mouse pluripotent cell lines which were obtained from David Suter (EPFL, Switzerland). They were also grown on gelatin plates using the same media as mESCs. To deplete OCT4 (ZHBTc4)[34,75], doxycycline was added to the growing media at the specified times (see figures and text).

Mouse embryonic fibroblasts (MEFs) were obtained from eviscerated torsos of E12-14 mouse embryos. Tissue from three embryos was minced with a scalpel on a plate with 2–3 mL of trypsin. The minced tissue was then placed in a 50 mL falcon tube containing 10 mL of trypsin for 10–15 min at 37 °C, followed by strong pipetting and

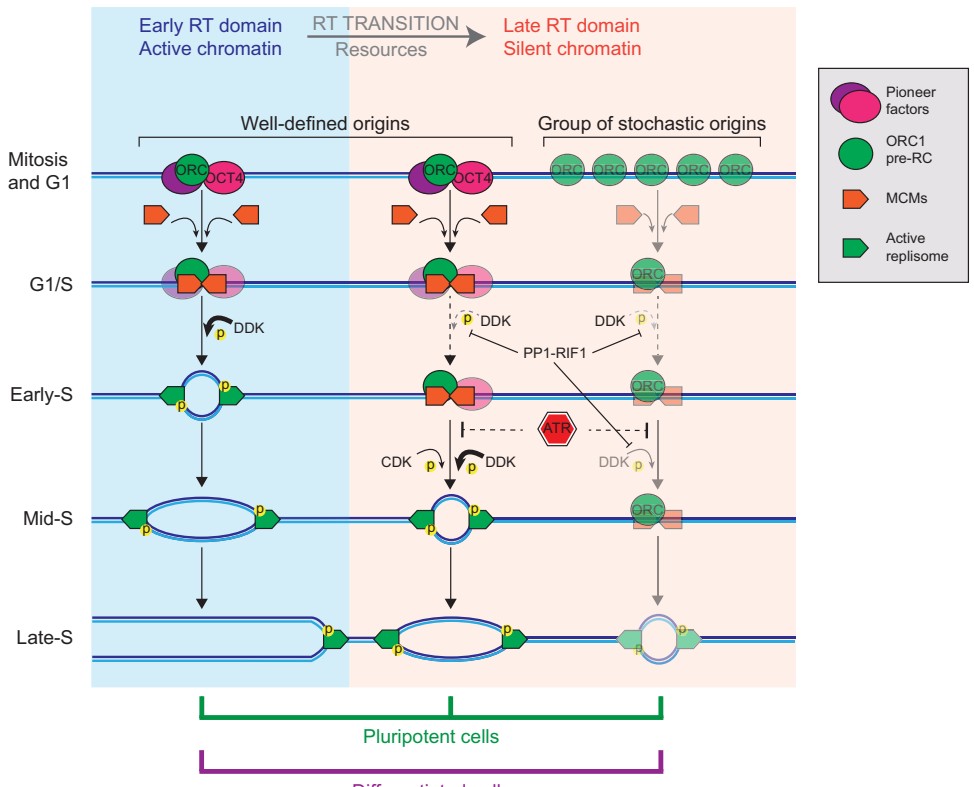

**Fig. 6 | Model for origin firing and RT.** Several layers control the efficiency and temporality of origin firing: locally, efficient origin firing is favoured by chromatin accessibility; globally, the transition into the late replicating program is controlled by ATR, RIF1 and the availability of resources. In mESCs, efficient, well-defined origins can be found in early, mid and late RT domains. In late RT domains, the firing of some of these origins depends on local recruitment of the pluripotency factor OCT4. In differentiated cells, efficient, well-defined origins can be found only in early and mid RT domains and replication of the late RT domains depends on low efficiency origins that cannot be mapped, when populations of cells are analysed.

another 10–15 min incubation at 37 °C. 20 mL of complete DMEM (DMEM, 10% FBS, P/S, L-glutamine) were added to the cell suspension, which was subsequently pipetted again and let to sit for 5 min. The supernatant was then recovered and centrifuged 5 min at $300 \times g$. The cell pellet was resuspended vigorously and plated on a 10 cm plate. This was considered passage 0 and from here MEFs were passaged, amplified, aliquoted and frozen. They were routinely cultured in complete DMEM and used for all experiments at passage three or four, except for the EU and γH2AX FACs experiments for which they were used at passages three and ten. Bone-marrow mouse mesenchymal stem cells (mMSCs) were a kind gift from Francesc Ventura (University of Barcelona, Spain). They had been obtained from the bone marrow of 8–10-week-old mice and were grown in complete DMEM with 15% FBS.

Depending on the experiments, cells were treated with the following reagents: nocodazole (100 ng/mL, Tocris 1228), aphidicolin (4 μM, Sigma-Aldrich A0781), thymidine (2 mM, Sigma-Aldrich T1895), RO-3306 (3 μM, Sigma-Aldrich SML0569), ATRi (100 nM, BAY-1895344, LubioScience S8666), hydroxyurea (100 μM or 2 mM, Sigma-Aldrich H8627), CDC7i (5 μM, in-house[42]) and TAK-931 (1 μM simurosertib, MedChem HY-100888).

All cells were routinely checked for mycoplasma contamination. All experiments were performed in agreement with the Swiss law on animal protection under license number GE11820.

### Synchronisation methods
Mitotic shake-off was conducted on mESCs and ZHBTc4 as follows. Three to four million cells were seeded in T225 flasks using 40 mL of media. Two days later, 100 ng/mL nocodazole was added to the media for 4–6 h. Media was then removed, and 15 mL of PBS were carefully added to the cells. Round cells were lifted by tapping the sides of the

flasks and visually inspected. Depending on the conditions of the experiment, cell suspensions from different flasks were strained through a 40 μm mesh and pooled into 50 mL tubes, centrifuged at room temperature ($400 \times g$, 5 min), washed once with 15 mL of PBS and centrifuged again. Pelleted cells were then carefully resuspended and plated. Commonly, the cells lifted from two T225 flasks would be seeded on one 10 cm plate (for EdU-seq and EU-seq) or on one 6 or 12 multi-well plate (for cell cycle analysis).

For thymidine or aphidicolin synchronisation, pluripotent cells (0.5–1 × 10^6 mESCs or ZHBTc4) or MEFs (1.5–2 × 10^6 cells) were seeded on 10 cm plates. 24–48 h later, 2 mM thymidine or 4 μM aphidicolin was added to the media for 10–12 h. Cells were washed three to four times with PBS and released into fresh media.

### Flow cytometry and cell cycle analysis
All pluripotent cell lines used (mESCs and ZHBTc4) were synchronised by mitotic shake-off, following, as needed, a doxycycline regime for the tet-off cell line (see figures and the results section for details). Cells lifted from two T225 flasks were plated on a gelatin 12-well multi-well plate. Cells were collected with trypsin at the indicated time points, with 50 μM EdU added during the last 30 min. After centrifugation, cell pellets were resuspended in PBS and fixed in methanol (final concentration 90%) and stored at −20 °C until further processing. The samples were centrifuged at $400 \times g$ for 10 min at 4 °C. The pellets were resuspended in 0.2% triton/PBS, transferred to a 96-well V-bottom plate and incubated at room temperature for 20 min. The plate was then centrifuged, supernatants removed by tilting and the sample wells washed; followed by another centrifugation, and supernatants removal. A click-IT cocktail mix was prepared containing, in one ml of final volume, 855 μL of 100 mM Tris-HCl pH 8, 100 μL of 1 M sodium

ascorbate, 40 μL of 100 mM CuSO4 and 1 μL of Alexa 647 Azide (Invitrogen A10277). Each sample/well was resuspended with 100–150 μL of the click-IT cocktail mix and incubated protected from light at room temperature for 30 min. The plate was then centrifuged, supernatants discarded, and pellets washed in PBS. After another centrifugation, pellets were resuspended in a solution containing 150 μL of PBS, 0.5 μg RNase (Roche, 11119915001) and 1 μg/mL of propidium iodide and incubated for at least 2 h at room temperature or at 37 °C. For EU quantification, asynchronous growing mESCs and MEFs were treated with 0.5 mM of EU (5-ethynil-uridine, Jena Biosciences, Cat. No. CLK-N002-10) for 30 min before collection by trypsin and fixed in 90% methanol/PBS. The cells were stored at −20 °C until further use. The fixed cells were pelleted (500 × g, 10 min at 4 °C), washed in cold PBS and pelleted again. The samples were then resuspended in fresh PBS and processed for EU imaging following with the Click-IT RNA imaging kit (Invitrogen C10330) but using the Alexa 647 Azide antibody (Invitrogen A10277) instead. The samples were washed in PBS and stained with FxCycle Violet Stain (F10347), fractionated as in the Repli-seq and EU-seq experiments and analysed in a Beckman Coulter Cytoflex device.

## Western Blot

Samples were collected with trypsin, pelleted and kept at −80 °C. They were resuspended in 100 μL of lysis buffer containing 10 mM Tris-HCl pH 7.5, 50 mM NaCl, 1 mM EDTA, 1 mM DTT, 10% glycerol, 10 mM Na-molybdate, protease inhibitors (EDTA free, Pierce, A32965) and phosphatase inhibitors (Thermo, 78420) and sonicated in a Diagenode Bioruptor waterbath in two cycles of 10 min (30 s on, 30 s off). The soluble fraction was recovered by centrifugation (10 min, 12,000 × g at 4 °C) and protein concentration was measured by the Bradford method (Biorad, 500-0006). Samples were then boiled in SDS Novex Buffer (Thermo LC2676) at 95 °C for 5 min. Between 20–40 μg of protein sample were loaded on each lane on 10–15% SDS-PAGE home-made gels. The wet transfer to nitrocellulose membranes (0.45 μm Porablot NCP-Macherey Nagel, 741280) took 90 min at a constant 250 A current in VWR tanks in transfer buffer (20% methanol, Tris, glycine). A control Ponceau staining was performed for 2–3 min and membranes were then subsequently washed in a big volume of deionized water. Membranes were blocked for one hour with 0.5% BSA (Roche, 10735086001) in TBST buffer. Primary antibodies were incubated overnight at 4 °C on a shaking platform. The following antibodies were used: γH2AX (1:1000, Upstate, cat. no. 05-636); p-CHK1 (1:1000, Cell signalling cat. no. 2341); PCNA (1:1000, Millipore cat. no. MABE288); OCT3/4 (1:500, Santa Cruz, cat. no. sc5279) and GAPDH (1:1000, Abcam, cat. no. ab181602). Membranes were washed three times in TBST and incubated in secondary antibodies-HRP (Pierce Goat anti mouse HRP cat. no. 31430 or goat anti-rabbit HRP, 31460) diluted 1:10000 in 2.5% BSA/TBST) for one hour on a shaking platform. After four TBST washes, membranes were revealed using ECL (Advantsa WITEC WesternBright, K12045-D20) and imaged in Amersham or Licor devices. Unprocessed scans of the blots are shown in Source Data.

## EdU-seq

Cells were treated with 40 μM of EdU for 30 min before collection by trypsin. EdU-seq was performed as in ref. 21 but with certain modifications. Methanol-fixed cells were pelleted, resuspended in 0.2% Triton and incubated for 20 min, followed by another centrifugation and one PBS wash. Cells were pelleted again, resuspended in 300–500 μL of click-IT solution and incubated for 30 min. Afterwards, cells were pelleted and washed, followed by lysis in 300 μL of buffer for a minimum of 3 h at 55 °C. Phenol/chloroform extraction and chloroform purification were sequentially carried in the same phase-lock tube. DNA was recovered by precipitation with NaCl and ethanol, quantified by Qubit broad range kit and kept frozen at −20 °C until later use or immediately processed. Between 1 and 5 μg of total DNA

were diluted in 105 μL of water and sonicated in a Diagenode Pico Bioruptor device for 7 cycles (15 s ON, 90 s OFF). DNA was then cleaned as specified in the abovementioned protocol using MyOne Streptavidin C1 Dynabeads and posteriorly eluted in 55 μL of 10 mM Tris pH 8 and 1.1 μL 2-mercaptoethanol. DNA library preparation and sequencing were performed at the Genomics platform of the University of Geneva. Sequenced samples are shown in Supplementary Data 1.

## Repli-seq

Cells were seeded in T225 flasks and grown for 48 h after which EdU (40 μM) was added for 30 min. At least 20 million cells were harvested per sample. The cells were collected by trypsinization, pelleted, resuspended in PBS, fixed in methanol added drop-wise (final concentration 90%) and stored at −20 °C until further processing. Cells were centrifuged at 4 °C, 400 G for 10 min, washed with 2–5 mL PBS, pelleted again and permeabilized by resuspending in 5 mL 2% triton/PBS for 30 min. Cells were pelleted, washed in PBS and passed into a 5 mL low bind tube. They were centrifuged, resuspended in 1–5 mL of Biotin-Azide Reaction cocktail (as in EdU-seq) and incubated for 30 min at room temperature. Cells were then centrifuged, washed once in PBS and resuspended in PBS containing 500 μg/mL propidium iodide and 500-1000 U of RNaseA (Thermo EN0531). The sample was incubated for 30 min at 37 °C and kept at 4 °C until the day after. The cells were then sorted in six fractions according to the cell cycle PI profile on a MOFLO Astrios EQ (Beckman Coulter) at the Cytometry platform of the University of Geneva. At least one million cells per fraction were sorted directly into low bind centrifuge tubes, which were later centrifuged and processed as in the EdU-seq protocol. Sequenced samples are shown in Supplementary Data 1.

## EU-seq

For FACS sorted EU-seq, one million mESCs or 3 million MEFs were seeded on 10 cm plates; the following day, EU (5-ethynil-uridine, Jena Biosciences, Cat. No. CLK-N002-10) was added at a concentration of 0.5 mM 30 min before collecting the cells by trypsinization and fixing them in methanol. The cells were stored at −20 °C overnight or longer. The fixed cells were pelleted (500 × g, 10 min at 4 °C), washed in cold PBS and pelleted again. The samples were then resuspended in fresh PBS, stained with FxCycle Violet Stain (F10347), fractionated as in the Repli-seq experiments, pelleted (600 × g, 15 min, 4 °C) and immediately resuspended in 600 μL Trizol. The samples were then processed as in ref. 63, but with certain modifications. Briefly, total RNA was extracted through chloroform purification and ethanol precipitation. 10–15 μg of EU-labelled RNA was processed with the Click-iT Nascent RNA Capture Kit (Invitrogen, Cat. No. C-10365), except that the biotin-azide from the kit was substituted with cleavable biotin-azide (Azide-PEG(3 + 3)-S-S-biotin) (Jena Biosciences, CLK-A2112-10). The click-IT reaction took place on a volume of 50 μL for 30 min at room temperature. RNA samples were then precipitated by adding 350 μL of water, 16 μL of 5 M NaCl, 1 μL of glycogen and 832 μL of cold ethanol, incubating the mix at −70 °C and centrifuging for 30–60 min at 4 °C, 16,000 × g. The RNA pellet was washed twice in a cold solution of 75% ethanol/water, centrifuged and the pellet resuspended in 400 μL of water. Biotinylated RNA was captured using 200 μL of Dynabeads MyOne streptavidin C1 (Invitrogen, 65001) per sample. The beads had been previously washed thrice in buffer W1x (5 mM Tris-HCl pH 7.5, 0.5 mM EDTA, 1 M NaCl, 0.5% Tween-20), twice in Solution A (0.1 M NaOH, 0.05 M NaCl in water) and twice in Solution B (0.1 M NaCl in water). The volume of solutions used for washing the beads was calculated as 200 μL per sample and scaled up depending on the total number of samples. After the last wash, the beads were removed from the magnet and resuspended in buffer W2x (same as W1x but two times concentrated) in two times the original volume (400 μL per sample). 400 μL of this washed bead mix were added to each biotinylated RNA sample and incubated 30 min at room temperature on a rotating

wheel. Afterwards, the bead-RNA samples were washed three times with 800 μL of buffer W1x. Each Biotinylated-RNA sample was eluted by resuspending the beads in a solution containing 55 μL of 10 mM Tris pH 8 and 1.1 μL of 2-mercaptoethanol and incubating them for one hour at room temperature. RNA libraries were generated without the ribodepletion step and sequenced at the Genomics platform of the University of Geneva. Sequenced samples, number of reads, their associated accession numbers are shown in Supplementary Data 1.

### Data processing

**Alignment.** FastQ files of the sequencing reads were aligned on the mouse reference genome NCBI Build GRCm38/mm10 using the Burrows-Wheeler Alignment tool v.0.7.17. The resulting SAM files were converted to BAM files keeping only the uniquely mapped reads. Samtools v1.9 was used to sort, remove the PCR duplicates and index the BAM file (https://arxiv.org/abs/1303.3997).

**BigWig tracks.** The processed BAM files have been transformed to coverage bigWig tracks using the bamCoverage function of deeptools v.3.5.1. The coverage was calculated as the normalized number of reads per bins per million mapped reads (BPM) with minimum mapping quality of 30 (https://doi.org/10.1093/nar/gkw257).

**Replication timing domains.** For the characterization of the replication timing domains of mESCs, MEFs and mMSC the processed BAM files of Repli-seq experiment were given as input to the Repliscan tool[102] using a window size of 10 kb. Replication domains are shown in Supplementary Data 1.

**IZ calling.** For the mapping of early IZs, we used an EdU-seq dataset from mESCs that were synchronized with nocodazole and MSO, released from the block in media containing EdU and hydroxyurea, and collected 2 h later (Fig. 1e, f: track (ix)). Mid and late RT IZs were mapped using an EdU-seq dataset from mESCs 8 h after release from a MSO; the cells were treated with a CDC7 inhibitor (5 μM, previously used by our group)[42] between 3 and 6 h after release from the MSO and with hydroxyurea for 1.5 h prior to harvesting (Fig. 1e, f: track (xi)). An in-house bash script was developed to call the origins in early, mid and late replicated domains (as identified from Repli-seq experiments). For the different samples, we divided the genome in early, mid and late domains according to the Repli-seq experiments. The chromosomes were split into 1 kb bins and the EdU-seq signal was extracted from bigWig tracks using bigWigAverageOverBed (UCSC https://hgdownload.cse.ucsc.edu/admin/exe/linux.x86_64/). Given the variability in the distribution of signal among different RT domains (higher in early, lower in late RT domains), signal cut-offs were empirically set at 50%, 25%, and 5% of the highest bins for early, mid, and late RT domains, respectively, with stricter thresholds applied in later domains due to lower peak-to-background ratios. From here, two refining steps were applied. First, broad regions were called after mm10 genome binning (1 kb) followed by signal extraction. Secondly, peaks were called after binning (1 kb) the broad regions called in the first refining step followed by signal extraction. Bins corresponding to the top 50% of the signal (median) were selected to define IZs. This method turned to be efficient for the identification of origins that tended to be clustered in short genomic areas, especially in the early-S replicating domains. Origins that overlapped with the genomic regions included in the ENCODE mm10 blacklist (regions with anomalous, unstructured, or high signal in next-generation sequencing experiments independent of cell line or experiment) were discarded as potential artifacts. The center of every origin was assessed by the refinepeak function of the macs3 algorithm using as input the coordinates of the called origins and the respective processed BAM file. The mid and late origins were shortlisted by discarding those that could potentially be confused with ongoing forks coming from earlier replicated domains.

Given the presence of different cell populations (due to the nature of the bulk experiments) and to avoid false positive calls, we discarded those mid and late origins that mapped <500 kb from neighbouring replication domains from which ongoing forks could come (for mid origins, early RT domains; for late origins, early and mid RT domains). The choice of 500 kb was decided after calculating the fork speed from the EdU-seq experiments; see Supplementary Fig. 2 (scheme on IZ calling), Fig. 2b, c (fork speed) and the paragraph on fork progression in this section. IZs detected on chromosome Y were discarded form further analysis.

To create a random dataset of control regions for the signal analysis of mESC IZs, we first generated 10 million random positions of one base pair size using the mouse reference genome mm10 (chr1-X). Subsequently, these positions were intersected with the regions originated from the extension of the refined early, mid and late RT IZs for 50 kb in both directions. The random positions that overlapped with the span of 50 kb downstream – refined IZ center – 50 kb upstream were discarded. Lastly, the non-overlapping random positions were shuffled and the first 1000 positions were selected as the final control dataset. IZ classification is shown in Supplementary Data 1.

**IZ S-phase plotting.** The EdU BPM normalized signal corresponding to the center of every mESC Early-Mid-Late RT IZ was extracted for the time points 2–12 h from the mESC EdUseq timecourse experiments. The signal was transformed to log2 + 1 scale to avoid the log transformation error of zero values associated with the absence of replication signal of the IZs at specific timepoints. Every dot depicts the signal in the center of an IZ. Red lines indicate the median values of every group.

**EdU-seq average signal.** The EdU normalized signal (BPM) around every IZ (500 kb upstream and downstream) was extracted at 1 kb bin resolution. The values of each IZ group (ES -early S-, MS -mid S-, and LS -late S-) were averaged per bin position. For every group of IZs the bin with the lowest averaged BPM value was considered as background level and was hence subtracted from the averages of every bin within the same group.

**Statistical comparison of histone signal.** To calculate the local background signal for each control region, as well as for every Early, Mid, and Late RT IZs, we used the mean signal of the examined histone marks at the outer edges of the plotted region (Fig. 4). Specifically, for each region, the first and last 100 kb from the plotted area were considered, using 1 kb bins, to determine the average local background signal. This background value was then subtracted from the signal of the central 1 kb bin of the respective region. Comparisons of signal distributions between groups were performed using the Wilcoxon rank-sum test.

**Fork progression.** To monitor fork progression, we used a subset of isolated IZs (see Supplementary Data 1), which were called from the first fraction (G1) of the Repli-seq mESC experiments. We defined isolated IZs as those located at least 100 kb away from any other IZ in the same Repli-seq dataset. For these isolated IZs, we plotted the average EdU-seq signal from all the available time-course experiments (as seen in the figures). The centre value was then refined to 10 kb resolution through the refinepeak function of macs3 and used to extract the distance the fork travelled between the time points.

**Residual regression analysis.** To assess whether OCT4 depletion causes systematic shifts in replication timing across the genome, we performed a residual-based linear regression analysis. The average normalized EdU signal was quantified in both control (OCT4-ON) and OCT4 depleted (OCT4-OFF) conditions across early (ES), mid (MS), and late (LS) RT initiation zones.

For each zone (ES, MS, LS), we computed the $\log_2$-transformed replication timing signals for both OCT4-ON and OCT4-OFF samples. Residual values were then defined as the difference between the $\log_2$-transformed OCT4-ON signal and the corresponding OCT4-OFF signal for each initiation zone (see Eq. (1)).

$$\text{Residual} = \log_2(\text{OCT4} - \text{ON}) - \log_2(\text{OCT4} - \text{OFF}) = \log_2\left(\frac{\text{OCT4} - \text{ON}}{\text{OCT4} - \text{OFF}}\right) \quad (1)$$

These residuals represent the degree of deviation in replication timing from the OCT4 depleted state toward the control. To test whether this deviation systematically depends on the OCT4 control state, we fit a linear regression model of the form (see Eq. (2)).

$$\text{Residual} \sim \log_2(\text{OCT4} - \text{ON}) \quad (2)$$

The null hypothesis tested was that the slope of this relationship is zero, i.e., that residuals are independent of the replication signal under the control condition. A significant non-zero slope would indicate that the extent of deviation from the OCT4-depleted varies systematically with the specific replication timing initiation zones in the OCT4-ON state, suggesting a locus-specific response to OCT4 loss.

**EU-seq analysis: transcript quantification and stratification.** For each EUseq fraction, transcript-level abundance was quantified as Transcripts Per Million (TPM) using TPMCalculator[103]. TPMCalculator was applied directly to alignment files (BAM files) with default parameters, using the *Mus musculus* mm10 genome and RefSeq curated transcript annotations in GTF format. Although originally developed for RNA-seq, TPMCalculator enables estimation of transcript abundance from any sequencing data aligned to a transcriptome or genome, including EUseq, by computing normalized expression values for annotated genomic features. Following TPM quantification, transcripts were ranked by TPM value and stratified into deciles (10% quantile bins). This stratification was used to assess transcriptional differences across EUseq fractions by comparing the relative abundance of transcripts within each quantile group. The results were hierarchically clustered according to the Euclidean distance.

**EU signal around TSS, TTS and enhancers.** A total of 137,039 enhancers from the 13 available mouse embryonic stem cell (mESC) lines included in the EnhancerAtlas database (v2.0)[68] were compiled and used to calculate the distance to the nearest enhancer from each of our defined early, mid, and late initiation zones (IZs). In parallel, all 35,513 canonical transcripts annotated in the mm10 reference curated genome (excluding transcripts located on chromosome Y) were used to determine the average distance of early, mid, and late RT IZs to their closest transcription start site (TSS) and transcription termination site (TTS). Distance calculations were performed using genomic coordinate-based proximity analysis.

### Reporting summary
Further information on research design is available in the Nature Portfolio Reporting Summary linked to this article.

## Data availability
All the data produced in this study has been deposited in have been deposited in NCBI's Gene Expression Omnibus with the GEO Series accession numbers: GSE271841, GSE271846 and GSE271847. Source data are provided with this paper.

## Code availability
The bash script for peak calling can be downloaded from GitHub (https://github.com/VSDionellis/OCT4_ESC_IZ_replication)[104].

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

## Acknowledgements

We are grateful to David Suter, Cédric Deluz and Francesc Ventura for sharing cell lines and giving us technical support. We would like to acknowledge the help of Maëlys Alemany on cytometry analysis and Samia Barriot, Christine Seguin and Laurence Tropia for technical support. We thank the members of the Halazonetis laboratory for helpful discussions and suggestions. We would like also to thank Nicolas Roggli for the production of the illustrations, as well as the Flow Cytometry (Jean-Pierre Aubry-Lachainaye and Grégory Schneiter) and Genomics platforms (Mylène Docquier, Brice Petit, Christelle Barraclough and Didier Chollet) of the University of Geneva. This study was supported by grants from the European Commission (788681, REPLISTRESS), the Swiss National Science Foundation (10006350) and the ACLON Foundation.

## Author contributions

Conceptualization and manuscript writing: E.R.-C. and T.D.H. Experimental work: E.R.-C., L.B., S.G.N. and E.K. Data processing: V.D. Analysis of results: E.R.-C., V.D. and T.D.H. Funding acquisition: T.D.H.

## Competing interests

The authors declare no competing interests.
