## [Transparent Peer Review file · Nature Communications]

OCT4 enhances the firing efficiency of late DNA replication origins in mouse embryonic stem cells

Corresponding Author: Professor Thanos Halazonetis

Version 0:

Reviewer comments:

Reviewer #1

(Remarks to the Author)

In this manuscript, Rodriguez-Carballo and coworkers studied replication initiation and timing in mouse embryonic stem cells (ESC), mesenchymal stem cells (MSC), and mouse embryonic fibroblasts (MEF) using combinations of inhibitors as well as a Tet-off cell model and genome-wide assays (EdU-seq, OK-seq, and EU-seq) performed under different regimes of cell synchronization. They set out to identify replication initiation zones (IZ) in ESC and categorize these into early-S, mid-S, and late-S IZ based on Repli-seq and EdU-seq using selected combinations of inhibitors to synchronize cells, limit replication fork progression, and firing of late replication origins. Their main claims are that ESC are distinct compared to MSC and MEF by having origin firing at late-S IZ already early in their S-phase, and that this is induced by the pluripotency factor Oct4.

The work presents an interesting hypothesis and a potential new role for pluripotency factors. While the overall technical quality and general handling of the genomic data is good (with the exception of the IZ calling and categorization, see below) and reflects solid experience with these methodologies, it is hard not to perceive the manuscript as premature and notice that the central claims are poorly supported by the presented evidence. Some parts appear meticulous and thoroughly thought through, e.g. the interrelation of Repli-seq and mitotic shake off synchronization EdU-seq in ESC, as well as the use of multiple strategies to perturb late replication origin firing in Figure 3. However, the general robustness of the presented evidence does not meet standards, as even the most critical experiments supporting the main claims rely on a single line of evidence only performed as a single replicate or rather circumstantial evidence. Moreover, central parts lack control experiments to support the claims and that the effects are not indirect consequences (e.g. FACS profiles of cell cycle progression to rule out effects are indirect consequences).

MAJOR POINTS

1) Initiation Zone calling and classification:

A pivoting point for the manuscript is the identification and classification of IZs in ESC. Despite its central position in the reasoning of the manuscript, the presented methodology is complex, obscure, and has fundamental shortcomings that need to be addressed.

a) Most critically, a large fraction of the mid-S and late-S IZ might be false positive and originate from progressing early-S forks. The sample 'k' used for identification of mid-S and late-S IZ, was generated after treatment with CDC7i followed by treatment with HU, and the EdU signal in this sample comes from not only replication initiation, but also from progressing replication forks. At the 6.5 h time point used for HU induction, replication has progressed considerably from early IZ as can be seen from samples 'b' to 'e'. Also, as illustrated in sample 'j', CDC7i leads to reduced firing from origins, so EdU-seq signal is derived only from replication forks initiated before the inhibition. This results in piled up signal at the dual flanks of earlier IZ, e.g. at app. 53.5 and 54.5 as well as 55.5 and 58.5 Mb in 'j' originating from IZ at 54 and 57.5 Mb which have visible replication already in the 4 h sample 'b'. Importantly, these flanks are also present in the sample 'k' additionally treated with HU and used for IZ peak calling. Thus, high signal in this sample comes not only from firing origins, but also from replication forks that were initiated much earlier and have progressed considerably. In other words, the profile of sample 'k' reveals not only areas of initiation, but also the regions where forks had progressed to after 3 h of CDC7 inhibition and the subsequent incubation. This is a fundamental confounding effect that questions the validity of the called IZs.

b) The peak calling and IZ classification process is complex and not sufficiently well presented in the current overview figure (Sup. Fig. 2a). This creates a lot of room for confusion, misunderstanding, and wasted reader time. Complex data processing is a problem for reproduction, interpretation, and transparency, so it is generally a sound strategy to keep things as simple as possible. (i) If the complex scheme is kept, this figure certainly needs to be revised to provide a more accurate and understandable presentation of the steps. Instead of combining sample treatment (which is specified elsewhere), data processing (which is generic and identical for all samples) and peak-calling, it should focus on clarifying the complex set of steps for the IZ calling and classification. It should also include specific information on which samples were used for which steps. Finally, the supplemental data sets and tables are on the light side and should ideally contain more information on the values used for the selection process. (ii) In its current version and in combination with Sup Fig 2b it is too easy to misinterpret the inclusion of sample 'e' as a part of the IZ calling and classification for mid-S IZ. It is unclear why the sample 'e' (EdU-seq from ESC 7 h after mitotic shake off synchronization) is included in the figure presenting the calling of initiation zones (Sup. Fig. 2b). From the methods and overview figure it does not seem to be relevant, so its inclusion in this figure opens for potential misinterpretations about its use in the IZ calling and classification. (iii) It is not clear how and why the seemingly arbitrary cutoffs of 50%, 25%, and 5% are implemented, and this should be explained and validated further. (iv) The authors should provide direct visualization of the IZ calling and classification similar to Sup. Fig. 2b, but of the central locus (chr8: 50-68 Mb) that is used for main Figure 1.

c) Once called, the IZ are seemingly classified solely based on the ESC Repli-seq data presented in Sup. Fig. 1c. As Repli-seq largely reflects fork progression rather than initiation, the distinction between mid-S and late-S IZ often relies on which stage that had the strongest fork progression in a given genomic region. For instance, replication initiating in the middle of the provided chr7 locus (at app 78Mb) occurs already in the S2 Repli-seq sample. As the authors rightly presents in the discussion and Figure 6, the central challenge is that replication is likely to be initiated from both well-defined origins that fire in most of the cell population and less well-defined 'stochastic origins' that fire in a minor part of the cell population. In the Repli-seq signal fork progression from well-defined origins will dominate, and a 'stochastic' origin giving rise to a mid-S or even early-S IZ with less signal and initiation would be overshadowed by late-S fork progression and therefore be classified as late-S based on the dominating Repli-seq signal. Thus, the current set of samples and IZ identification and classification approach has limited capability to discriminate initiation occurring in mid-S from that occurring in late-S. This is particularly problematic given the emphasis on late-S IZ in the later part of the manuscript. Perceivably, the solidity of the manuscript would be higher if the authors chose to only discriminate based on direct data (i.e. samples 'l' and 'k') rather than indirect inferences. As acknowledged by the authors, the late-S IZ might as well represent 'stochastic' or low-abundance origins, and this possibility and the procedure limitations should at least be explicit when the origin classification is introduced in the manuscript. If authors decide to proceed with the distinction between mid-S and late-S IZ, they should be introduced as 'putative', 'IZ at mid-S replicated DNA', or similar to more accurately represent the tentative interpretation of these regions.

2) Comparison between ESC, MSC, and MEF:

A central claim in the manuscript is that ESC are distinct from the other two studied cell lines by having replication initiating very early on in the S-phase at late-S IZ. This claim is solely based on Figure 1g, and this regrettably has several inbuilt assumptions that are not assessed at all. These assumptions need to be addressed as described in the following.

a) First and foremost, the mESC cells underwent additional treatment with HU, which lead to accumulation of signal in certain regions including the late-S IZ. This biases the comparison towards a stronger signal at these weak IZ in the ESC compared to the other cell types. The authors first need to correct this apple-to-banana comparison by presenting data from similar treatments across the cell lines, which in this case also includes accounting for differences in cell cycle progression (see below).

b) The manuscript omits FACS profiles of the nuclear content to confirm that these cells are indeed G1/S synchronized as the authors claim (line 148) - and fundamentally are comparable at all. It is very possible that the cells are in distinct parts of the cell cycle and not comparable at all when used for EdU-seq (more below).

c) The use of a single pulse of aphidicolin to G1/S synchronize ESC, MSC, and MEF in a comparable manner is at best unconventional. Papers from more conventional model systems for cell cycle studies, such as HeLa cells, indicate that aphidicolin leads to arrest both in G1 and at any part of the S-phase that the cell has made it to (e.g. Pedrali-Noy et al, 1980, PMID: 6775308). While the aphidicolin treatment likely leads to a block, it can hardly be called a synchronization and certainly not 'at the G1/S-boundary' (line 148) without a thorough validation of this claim. Authors should use a more appropriate method to synchronize cells at the G1/S boundary, e.g. Palbociclib, and include FACS profiles from these very samples to increase the strength of the data. Ultimately, without a convincing methodology and validation of the underlying assumptions, the authors cannot claim that the difference in the signal at late-S IZ between the cell lines is a biological difference and not an indirect effect of differences in the cell cycle of these cells.

d) Given the centrality of this claim, it should be backed by an independent line of evidence, e.g. qPCR-based assays from the regions of interests.

e) The authors only provide EdU-seq data from early time points after aphidicolin synchronization in Figure 1g. Given the differing kinetics of the cell cycle between the cell types used, it is possible that the signal at the late-S IZ would also occur in MSC and MEF if later time points were included. To claim that this is a distinct feature for ESC, then the authors need to rule out that origins within these late-S IZ are firing later in the cell cycle of MSC and MEF.

3) Effects of Oct4 depletion:

a) Oct4 depletion is known to lead to a range of effects, such as cell cycle perturbation and differentiation of ESC, also for the ZHBTc4 model system used (PMID: 19968627; PMID: 14990861; PMID: 25324523). Thus, the reduced EdU-seq signal in Oct4-depleted cells at late-S IZ could be due to an indirect consequence of differentiation. (i) To maintain the claim that Oct4 stimulates origin firing at late-S IZ, the authors need to demonstrate that either no differentiation takes place or that the reported effect occurs in cells that are undifferentiated (e.g. through FACS sorting for Nanog levels). (ii) As it is challenging to get direct evidence of Oct4 binding and inducing initiation at late origins (and likely not a fair demand), authors should more clearly acknowledge the possibility of the relationship being indirect.

b) The evidence provided in Figure 5 is in its current form circumstantial. The authors have to provide actual data on Oct4 from the studied model system, a direct analysis of the Oct4 signal at the late-S IZ, and compare local Oct4 levels in the ZHBTc4 cells (ChIP-seq, ChIP qPCR, or similar). If the presented model is right, then Oct4 levels should be higher in the most affected late-S IZ (red dots in Figure 5e right panel) compared to the less affected late-S IZ. The comparison to public data from a distinct cell line in Figure 5c is not providing evidence suitable for testing their hypothesis.

4) Editing, communication, diligence, and figures:

The manuscript editing seems premature in several ways and needs adaptations to cater for a broad audience.

a) Rationales for using different inhibitor regimes are in some cases presented much after the initial use of the inhibitors. This is particularly clear for CDC7-inhibition, which is presented in Figure 1, but the rationale only follows in the results section describing Figure 3. Presumably, authors did a major restructuring of the manuscript late in the process without ensuring that information is presented in a logical order. Also, CDC7 inhibition is listed as CDC7i, TAK-931, and TAK in figures and the result section, and authors need to clarify if these are the same treatments and if so, make the text consistent. Finally, methods are lacking when it comes to description of the inhibitor treatments, and there is allegedly no description of the concentrations used when inhibiting CDC7. Relatedly, concentrations of e.g. HU are also inconsistently listed or left out in the manuscript and figures.

b) In figure 1 and Supplementary Figures 1 and 2 the authors present a complex set of combined treatments labelled 'a' to 'n'. While it is praiseworthy to use the small overview timelines (generally in many of the figures) as well as a 'unique identifier' for the many conditions, the later is not forthcoming for the readers' ability to digest the panels and findings. The current version of Figure 1 and supplement almost feels like a cryptography exercise where dozens of cross-references need to be made. The authors should help the readers further by using more informative labels that also describe the characteristics of the samples. This can certainly be combined with the use of a unique identifier, but it would enhance clarity to use one that is distinct from the lettering used to label the panels.

c) The description of the data visualization done in Figures 3-5 in methods is absent, and the bash script for peak calling is not accessible in the provided link (link broken, 404 error).

d) In Figure 1 and the corresponding supplement, the authors present the foundation for the central claims made in the manuscript. The work however only presents EdU-seq profiles from a single locus at chr8, and much of the conclusions are based on enrichment profiles a two alleged late-S IZ. Key conclusions in these figures should be substantiated by generalized visualizations like the heatmaps, average signal, and scatter plots presented in Figures 2-5.

OTHER POINTS

1) The title of the manuscript is misleading. Since the manuscript solely focuses on one of the pluripotency factors, Oct4, the title should reflect this. Generalization to 'pluripotency factors' is a stretch.

2) In the abstract, the authors claim that the spatiotemporal program of replication is not well-defined (line 29). This seems inappropriate given the solid amount of work on this, and it is also contradicted by the authors' own statements later in the manuscript, e.g. in lines 56-59!

3) The knowledge gap in the introduction could be described more clearly and thoroughly.

4) There are oddities in the EdU-seq signal profile of several of the samples. The authors should acknowledge this in the manuscript and explain those. This includes the enrichment in G1 samples in Sup Fig 1c, elevated signal 2 h after synchronization at Late IZs in Sup Fig 2c, abundant signal at late replicating regions in Figure 1c sample 'a' (e.g. in the middle of the chr8 locus), abundant signal in late replicating regions in the MSC 0.5h sample in Figure 1g (e.g. at the start of the locus).

5) EdU-seq and EU-seq normalization seems inconsistently mentioned. In some figures and in the IZ calling section of the methods 'BPM' is mentioned, but in other figures these labels are absent. There is a mentioning of log₂+1 transformation (line 531, but it is unclear which plots where the signal is log-transformed, and it is preferable to have such information in the relevant figure(s).

6) The overall quality of Figure 4 is low. This needs to be addressed. Specifically, Figure 4a culture dish, 4c and 4d heatmaps needs to be updated with higher quality versions.

7) The clustering done in Figure 4c is unsuited to conclude that "Of the early-replicating genes, more were transcriptionally active in mESC than in MEFs". Dedicated analyses of this should be made to support this claim, and a quantitative approach would convey this better. For example, a box or violin plot visualization of nascent RNA levels of each RT domains would be nice.

8) Stars (*) in Sup Fig 2b are not explained.

(Remarks on code availability)

The code link is broken, so a review could not be done.

Reviewer #2

(Remarks to the Author)

(Remarks on code availability)

Reviewer #3

(Remarks to the Author)

In this manuscript by Rodriguez-Carballo and colleagues, the authors show that mESCs fire mid to late initiation domains early when compared to MEFs or mesenchymal stem cells. Interestingly, this change of replication timing is mediated by pluripotency factors like OCT4 and correlates with chromatin accessibility. This is an interesting observation but the mechanisms by which the pluripotency factors are driving this change of replication specifically in mESCs are not elucidated. Below are the main comments which should clarify some of the outstanding questions:

1. What percentage of mid-late IZs are firing in ESCs? the authors should have a cumulative plots showing it in the main figure.
2. Fig1F, after CDC7i treatments, the overall origin firing seems to be diminished (e vs j). Could the reduction of mid and late IZ firing just be due to that?
3. Can the firing of mid and late IZs be a result of synchronization methods which induce replication stress for example use of high HU doses which completely stalls the forks which in turn activates origins locally in mESCs specifically? Can the authors dissect that out?
4. The authors show in Fig 4 that that the mid and late IZs are marked by open chromatin (not necessarily co-relating with transcription). However, this chromatin accessibility at mid and late IZs seems to be dependent on OCT4. How do the authors envisage OCT4 mediating chromatin accessibility? This mechanism needs to be dissected out as pluripotency factors driving changes of replication timing of mid and late IZs is the most interesting part of the entire manuscript.

(Remarks on code availability)

Reviewer #4

(Remarks to the Author)

(Remarks on code availability)

Reviewer #5

(Remarks to the Author)

In Rodriguez-Carballo et al., the authors thoroughly characterize DNA replication initiation throughout early-, middle- and late-S-phase and explore its relationships with experimental perturbations and broader chromatin organization. The authors specifically focused on mouse embryonic stem cells (mESCs) for most of their experiments as the short G1-phase of mESCs allowed for the authors to perform cell cycle sorting by capturing cells in 2hr intervals after mitotic exit. This sorting procedure critically enabled the authors to measure initiation, and ensuing replication fork progression, with EdU-seq protocols which provide higher resolution than competing Repli-seq and OK-seq assays. After characterizing initiation zones (IZs) mESCs, mMSCs, and MEFs, the authors began to probe specific relationships between replication timing (RT) and IZs through

perturbations such as CDC7 inhibition (via TAK-931), dNTP depletion, ATR inhibition, and CDK1 inhibition. The authors comprehensively rationalize why each experiment was performed and relate their results to broader literature & prevailing theories on the role of different checkpoints (i.e. Fig 3c).

In the second half of the paper, the authors transitioned to examining how replication varies between epigenetic and transcriptional states. The authors make valuable use of external ATAC-seq and other epigenetic data to support their claim that open chromatin features are present at IZs. Finally, the paper concludes with an investigation into how pluripotency factor binding is enriched across all IZs and that depletion of OCT4, through a dox-inducible system, led to decreased chromatin accessibility and origin firing efficiency at late IZs.

Overall, the data and biological insights from this paper represent a meaningful contribution to the field which will spur future investigation into how late-firing origins differ from early firing ones and their relationships with epigenetic modifiers such as OCT4 and other pluripotency factors. In spite of my general enthusiasm, there were several areas in the paper which could be strengthened through more rigorous statistical analysis. I would particularly like these comments to be addressed before recommending publication.

Major comments

1. L127-152: Results from Fig 1c, f, g are mostly limited to arrows pointing at very small peaks within a ~15Mb region of chr8. These results would be significantly strengthened, and easier to interpret, if the authors added an additional panel(s) to Fig 1 which demonstrates that these trends are conserved across the entire genome (perhaps something like Figs 2b, 2g, 5e?). Looking genome-wide would also enable the authors to strengthen their claims by testing for statistical significance, effect sizes, etc.
2. L217-224: The clustering approach used by the authors is not statistically rigorous enough to claim that certain RT domains are enriched for certain transcription levels (L219-221) or that early-replicating genes were more transcriptionally active in mESCs than in MEFs (L221-223). The clustering approach might be okay for visualization purposes in Fig 4c but this is another result that would be strengthened by statistical hypothesis testing and an additional figure panel that more clearly supports the claim on L221-223.
3. L236-237: Another example where statistical testing would better show that there is still enrichment of epigenetic marks AND that effect size of this enrichment is lower in mid/late IZs compared to early IZs.
4. L258-269: I like the fact that the authors performed genome-wide analysis via scatterplot in Figs 5e and S6d,e; however, I would prefer if the authors reported p-values to compare "observed vs expected" rather than the $y=mx+b$ linear regression coefficients and the angle between linear regression fit of the data vs expected. One suggestion would be to compute the residual values between the observed data points and expected values → fit a linear regression model to the residuals → report a p-value for a hypothesis test whose null hypothesis is that the slope is zero. If using python, this can be easily done using the p-values reported by scipy's linear regression tool
<https://docs.scipy.org/doc/scipy/reference/generated/scipy.stats.linregress.html>
5. Change "pluripotency factors" to "OCT4" in the title or replace/remove the "pluripotency factors enhance the" clause all together. While the authors revealed that all four pluripotency factors (OCT4, SOX2, KLF4, NANOG) are enriched at early, mid and late IZs in Fig 4h, only OCT4 was further investigated in Fig 5 to show that its absence delayed the firing of mid- and late IZs.

Minor comments

1. L65-73: Consider citing Emerson et al., Nature, 2022 somewhere in this introduction paragraph as they seem to have investigated late IZs in human cells, albeit at 50kb resolution since they used RepliSeq.
2. L697-702: Define the x-axis for Fig S1a. I think it's ploidy but should be specified in the legend and axis label.
3. L105: Specify the resolution of the RT profiles you obtain here. You mention that they're 10kb in the methods (L490) but this seems to be an important point as you mention the high-resolution of this data as a key differentiating factor from previous Repli-seq studies of late IZs. (lines 65-73).
4. L108: "Repli-seq experiments extrapolate time from genomic content" is a confusing phrase. Please reword.
5. L108-119: Split the heatmap in Fig S1e into a new panel S1f (keeping the two scheme diagrams in S1e) so that you can separately reference experimental design vs heatmap results in the main text. Also, only show the upper left or lower right triangle of this heatmap since it is symmetric. Specify the type of correlation used in colorbar or legend.
6. L130-152: Instead of assigning letters for each track in Figs 1b-f, assign roman numerals or some alternative scheme. Having letters for both tracks and figure panels can be confusing when reading the main text.
7. L249-254: Reference Fig 5b somewhere near here as mention of Fig 5b seems to be missing from the main text.
8. L550-553: More code used to analyze data and produce figures should be published on github. The authors released

code for peak calling; however, code to produce Fig 2b,c,g, 4c,d, 5e, etc. should also be released.

(Remarks on code availability)

Only one script for peak calling is provided. The authors could also include other code useful for reproducing the results, for instance the EdU-seq or fork progression code.

Version 1:

Reviewer comments:

Reviewer #1

(Remarks to the Author)

In this revised manuscript, the authors have addressed some of our previously presented points while leaving central points unaddressed or only partially addressed.

It is very possible that mESC differ from MEFs and MSCs in their replication timing regulation and that Oct4 accelerates local replication as the authors claim. This would not be a surprising finding given the existing knowledge in the field.

However, without the right evidence it is not possible to discriminate between this and other models, and we find that the overall methodology does not provide the required rigor.

This undermines the strong conclusions made in the paper, and in its current state several of the main points presented in the abstract are not substantiated. At the same time the authors tend to drive a preferred interpretation and not entertain alternative explanations. We find that the combination of the one-sided reporting and the extent of the methodological weakness to be material, and the revised manuscript provides little new evidence to reduce our concerns.

We do understand and acknowledge that what the authors attempt to do is not technically trivial and that there are limitations on what the different cell types allow in terms of synchronization and assaying.

Given these limitations and inability/unwillingness to provide substantiating evidence it would be appropriate to present data with a more open reporting style that acknowledges limitations and alternative explanations more explicitly.

A bare minimum of diligence would require authors to tone down claims in concordance with the points presented below and to provide a transparent reporting on these confounding factors – both when results are presented and discussed and in a separate Study Limitations section.

Main unresolved points:

1. The authors overconclude and oversell in the abstract. This either demonstrates a disqualifying lack of care or a drive to push their claims beyond what the data supports. Main claims in the abstract were not and are still not supported by data, and in the absence of supporting data, the abstract needs to be modified to reflect actual findings. E.g.:

a. The statement that “we identified DNA replication initiation zones (IZs) in mESCs, mesenchymal stem cells and mouse embryo fibroblasts (MEFs)” is factually wrong. The authors only identify IZ from mESCs and then study EdU-signal levels at these sites in the other cell types. It is concerning that such simple matters are not congruently presented.

b. Similarly, as the authors acknowledge in their point-by-point response then they only investigate Oct4, yet generalize to pluripotency factors in the abstract: “This change in the replication timing program, which did not lead to shortening of the length of S phase, was driven by pluripotency transcription factors, notably OCT4...”.

c. The statement that “Uniquely in mESCs, we observed, within 1-2 hours of entry into S phase, origin firing at IZs that mapped to the mid and late-replicating genomic domains” is based on apple-to-banana comparisons. This was pointed out in our previous set of comments, and the problem is aggravated when the authors end up using the resultant absence of evidence to conclude evidence of absence. There are central dissimilarities in both data and methods: Firstly, the assays used to observe initiation in the cell types differ on central points and only mESC were treated with HU, whereas none of the tracks from the MEF and MSC had such a treatment. Secondly, IZs were only ever identified in mESC, and the replication in MEFs and MSCs was monitored as these sites. This results in a lopsided comparison where selection biases may affect conclusiveness. To conclude that this is a unique trait for mESC authors need to identify IZs in MEFs and MSCs using an identical approach and observe a lack at these sites. Without data that allows a more direct comparison, the statement should be softened or left out.

d. The use of ‘change’ in the statement “This change in the replication timing program” implies a that a comparison is performed across two related states and not across three distinct cell types. Moreover, given the limitations described above, this difference might as well be due to methodological differences.

2. In their response, the authors claim that the effect of Oct4 on cell cycle in ES cells is negligible as they assay cells 19-25 h

after induction, while proliferation decrease occurs at 48h and 72h in the previous work with the title “Oct-4 controls cell-cycle progression of embryonic stem cells” (PMID: 19968627). However, at 48 h the published work reports less than one third the number of control cells (PMID: 19968627, Fig 1B). It is inconceivable that the massive preceding reduction in proliferation occurred in the final hour of the assay or that it was not accompanied by perturbations of the cell cycle long before this time point.

The authors should with relative ease be able to demonstrate whether this is the explanation for their observations but chose not to provide such data for the revised manuscript. This should be provided.

3. Readers may get the impression that e.g. the ‘Late IZ’ are firing late. The authors now do clarify that this is not their claim. And it would be appropriate to make this clearer throughout the manuscript, e.g. by also clarifying this early in the reporting. The recurrent use of e.g. ‘late IZ’ throughout the manuscript e.g. lines 230, 275, 326, 327, 346 without this context gives the impression that the authors believe that initiation occurred at the indicated stage. The authors should throughout the manuscript make sure that initiation timing and initiation at loci predominantly replicating at a given stage is not conflated. Although wordier, one way would be to use the term ‘IZ in late replicating regions’ or ‘predominantly late replicating IZ’.

4. We understand the response provided regarding the IZ calling and agree at the data provided in the response figure reduces the concerns at the illustrated locus. However, the challenge of using algorithms and pipelines to search the entire genome, is that even in the best setup noisy data and edge cases tend to lead to false positives and negatives. This possibility is not handled well in the current version of the manuscript, and the authors have no figures or analyses that on a genome-wide scale address the robustness of the called IZs. Neither do they seek to use independent methodology (such as qPCR, which they describe as unsuitable in the response). Thus, conclusions end up relying on one line of evidence without any potential validation.

a. The authors should discuss (e.g. in discussion or study limitations) how both false positive and negative calls will affect conclusions.

b. Moreover, they should seek to validate the three categories of IZs, e.g. using the provided OK-seq data – conceivably IZs would have a step change in OK-seq directionality. This can and should be analyzed on a genomic level, and it would be helpful for the reader to know that e.g. 70% of the IZ in late replicating regions were associated with a change in OK-seq directionality.

c. Finally, the 500 kbp zone used for exclusion seems tight. In the provided examples this seems ok although the shoulders of the progressing fork signal in sample (xi) seems to be within the 500 kbp zone. For more jagged/noisy signal, it is possible that peaks would be called from these flanks just outside the zone. This might also not be the case, but the reader needs figures that allows her/him to judge this themselves, and currently it is hard to make a general assessment. The authors should therefore provide supplementary figures that illustrate and substantiates that this is an appropriate exclusion zone. They should also plot the density of late S IZ as a distance of mid S / early S IZ (and the same for mid S IZ relative to early S IZ). If the exclusion zone is appropriate, there should expectedly a rather flat distribution of IZs outside the 500 kbp zone.

5. mESC are heterogenous and widely known to spontaneously differentiate, and this is very pronounced when grown in serum (as done in this manuscript), while newer culture conditions relying on 2i medium reduces this. Therefore, the observed differences between mESC and MEFs / MSCs could be due to a more heterogenous cell population. With the used IZ classification, IZs end up being determined based on the RT profile in the dominant cell population. The manuscript should more directly acknowledge this, e.g. near the discussion in lines 300-303.

Minor points:

6. In the response the authors write: “we could hypothesize that because ESCs do not manage to replicate completely the whole genome in a given cell cycle, some late regions remain unreplicated. “ as a possible explanation for some of the aberrant profiles. This would be a huge controversial finding on its own, and it is plausible that other more mundane explanations exist, e.g. impurities in the assayed fraction. The aberrant tracks should be mentioned in the manuscript along with possible explanations.

7. We did not ask for exhaustive naming or complex names such as ‘MSO-Harv7h-CDC7i3-6-HU6.5-7’, just labeling. If the authors disagree on the need for rememberable and understandable labels, then at least restructure the layout of figure 1, so that the scheme in e is on the left side of the tracks in f (and similar for b, c). This structure is already used in other figures.

8. Suppl. Figure 2a contains BMP instead of BPM a couple of occasions.

(Remarks on code availability)

Reviewer #2

(Remarks to the Author)

I co-reviewed this manuscript with one of the reviewers who provided the listed reports. This is part of the Nature Communications initiative to facilitate training in peer review and to provide appropriate recognition for Early Career

Researchers who co-review manuscripts.

(Remarks on code availability)

Reviewer #3

(Remarks to the Author)

The authors have addressed all my concerns sufficiently and I can recommend the manuscript for publication.

(Remarks on code availability)

Reviewer #4

(Remarks to the Author)

(Remarks on code availability)

The authors used code / bioinformatics tools from other studies to produce the most of the figures from their manuscript. The original code from this manuscript pertains to peak calling which has been deposited into a github repository. This repository could be improved through the addition of a README.md file which specifies the R package dependencies and instructions for running their scripts.

Reviewer #5

(Remarks to the Author)

The authors have addressed my concerns.

(Remarks on code availability)

The provided code only includes a peak calling script. It would be more optimal to also include a .Rmd or other code file that shows how the authors created the figures.

Version 2:

Reviewer comments:

Reviewer #1

(Remarks to the Author)

The authors have generally performed revisions that address previously presented points.

This includes a rewritten and substantially changed abstract, where the major claims now better reflect the conclusiveness of the data.

In addition, the authors have included several passages that present and consider alternative interpretations.

In general, I can support publication.

A few minor points remain, but these can be addressed without further reviewing:

The title of the final section of results (line 275) still states "Pluripotency factors act as pioneer factors for mid and late IZs", as they do not investigate role of any other pluripotency factor other than OCT4 in this section, it should be updated to 'The pluripotency factor Oct4' or similar.

Given the centrality of Oct4 for the claims made in the manuscript, it would be appropriate to include a deeper discussion of the possibility of Oct4 having other effects on cell cycle progression including the points made in their response to the previous major point 2.

Reviewer #2

(Remarks to the Author)

RESPONSE TO THE COMMENTS OF THE REVIEWERS

Reviewer #1 (Remarks to the Author):

In this manuscript, Rodriguez-Carballo and coworkers studied replication initiation and timing in mouse embryonic stem cells (ESC), mesenchymal stem cells (MSC), and mouse embryonic fibroblasts (MEF) using combinations of inhibitors as well as a Tet-off cell model and genome-wide assays (EdU-seq, OK-seq, and EU-seq) performed under different regimes of cell synchronization. They set out to identify replication initiation zones (IZ) in ESC and categorize these into early-S, mid-S, and late-S IZ based on Repli-seq and EdU-seq using selected combinations of inhibitors to synchronize cells, limit replication fork progression, and firing of late replication origins. Their main claims are that ESC are distinct compared to MSC and MEF by having origin firing at late-S IZ already early in their S-phase, and that this is induced by the pluripotency factor Oct4.

The work presents an interesting hypothesis and a potential new role for pluripotency factors. While the overall technical quality and general handling of the genomic data is good (with the exception of the IZ calling and categorization, see below) and reflects solid experience with these methodologies, it is hard not to perceive the manuscript as premature and notice that the central claims are poorly supported by the presented evidence. Some parts appear meticulous and thoroughly thought through, e.g. the interrelation of Repli-seq and mitotic shake off synchronization EdU-seq in ESC, as well as the use of multiple strategies to perturb late replication origin firing in Figure 3. However, the general robustness of the presented evidence does not meet standards, as even the most critical experiments supporting the main claims rely on a single line of evidence only performed as a single replicate or rather circumstantial evidence. Moreover, central parts lack control experiments to support the claims and that the effects are not indirect consequences (e.g. FACS profiles of cell cycle progression to rule out effects are indirect consequences).

We thank the Reviewer for the above summary of her/his comments. As these comments are expanded below, we address them below.

MAJOR POINTS

1) Initiation Zone calling and classification:

A pivoting point for the manuscript is the identification and classification of IZs in ESC. Despite its central position in the reasoning of the manuscript, the presented methodology is complex, obscure, and has fundamental shortcomings that need to be addressed.

a) Most critically, a large fraction of the mid-S and late-S IZ might be false positive and originate from progressing early-S forks. The sample 'k' used for identification of mid-S and late-S IZ, was generated after treatment with CDC7i followed by treatment with HU, and the EdU signal in this sample comes from not only replication initiation, but also from progressing replication forks. At the 6.5 h time point used for HU induction, replication has progressed considerably from early IZ as can be seen from samples 'b' to 'e'. Also, as illustrated in sample 'j', CDC7i leads to reduced firing from origins, so EdU-seq signal is derived only from replication forks initiated before the inhibition. This results in piled up

signal at the dual flanks of earlier IZ, e.g. at app. 53.5 and 54.5 as well as 55.5 and 58.5 Mb in 'j' originating from IZ at 54 and 57.5 Mb which have visible replication already in the 4 h sample 'b'. Importantly, these flanks are also present in the sample 'k' additionally treated with HU and used for IZ peak calling. Thus, high signal in this sample comes not only from firing origins, but also from replication forks that were initiated much earlier and have progressed considerably. In other words, the profile of sample 'k' reveals not only areas of initiation, but also the regions where forks had progressed to after 3 h of CDC7 inhibition and the subsequent incubation. This is a fundamental confounding effect that questions the validity of the called IZs.

Of course the Reviewer is right that samples collected 7 h after mitotic exit or later will have signals corresponding to progressing forks and from newly-fired initiation zones (IZs). To correctly classify peaks as progressing forks or newly-fired IZs, we collected samples at multiple time points after mitotic exit (2, 4, 5, 6, 7, 8, 10, 12 h) and used several synchronization methods (thymidine, aphidicolin or mitotic exit with and without CDC7 inhibitor treatment). For example, at 4 h after mitotic exit, we observed firing of IZs (green and orange arrows in panel Fig. 1c (ii/b) below). At 5, 6 and 7 h after mitotic exit, we observed bidirectional progressing forks moving away from each other (red arrows in panels Fig. 1c (iii,iv,v/c,d,e) below). Thus, we can unambiguously distinguish IZs from progressing forks, since only the former lead to two symmetrically located peaks that move further apart from each other as time progresses.

We note that IZs do not fire at exactly the same time in all cells. This explains the continuous signal between the two progressing forks starting from the same IZ, as seen in Fig. 1c (v/e). When we added a CDC7 inhibitor (CDC7i) 3 h after mitotic exit, then all the signal comes from IZs that fired before the CDC7i was added to the cells. In this case, the two progressing forks are clearly distinguished from each other (Fig. 1f (x/j) below). In the next experiment, CDC7i was added at 3 h after mitotic exit and removed 3 h later, to allow the IZs that had not fired early (within the first 3 h after mitotic exit) to fire. In this experiment, we could clearly visualize the IZs (compare Fig. 1f (xi/k) with Fig. 1f (x/j) below) and, thus, we can distinguish newly-fired IZs from progressing forks.

Fig. 1 for Reviewers. Panels indicating origin firing at IZs (green and orange arrows for IZs at mid-S and late-S replicating genomic domains, respectively) and progressing forks (red arrows). Mouse ESCs were treated as described. These panels are taken from Fig. 1 of the manuscript. The panels were numbered a-n in the original version of the manuscript and i-xiv in the revised version. Both numberings are shown here. The IZ at 54 Mb and bidirectional progressing forks at 53.5 and 54.5 Mb are marked in panels Fig. 1c (ii/b) and Fig. 1f (x/j), respectively.

b) The peak calling and IZ classification process is complex and not sufficiently well presented in the current overview figure (Sup. Fig. 2a). This creates a lot of room for confusion, misunderstanding, and wasted reader time. Complex data processing is a problem for reproduction, interpretation, and transparency, so it is generally a sound strategy to keep things as simple as possible. (i) If the complex scheme is kept, this figure certainly needs to be revised to provide a more accurate and understandable presentation of the steps. Instead of combining sample treatment (which is specified elsewhere), data processing (which is generic and identical for all samples) and peak-calling, it should focus on clarifying the complex set of steps for the IZ calling and classification. It should also include specific information on which samples were used for which steps. Finally, the supplemental data sets and tables are on the light side and should ideally contain more information on the values used for the selection process. (ii) In its current version and in combination with Sup Fig 2b it is too easy to misinterpret the inclusion of sample 'e' as a part of the IZ calling and classification for mid-S IZ. It is unclear why the sample 'e' (EdU-seq from ESC 7 h after mitotic shake off synchronization) is included in the figure presenting the calling of initiation zones (Sup. Fig. 2b). From the methods and overview figure it does not seem to be relevant, so its inclusion in this figure opens for potential misinterpretations about its use in the IZ calling and classification. (iii) It is not clear how and why the seemingly arbitrary cutoffs of 50%, 25%, and 5% are implemented, and this should be explained and validated further. (iv) The authors should provide direct visualization of the IZ calling and classification similar to Sup. Fig. 2b, but of the central locus (chr8: 50-66 Mb) that is used for main Figure 1.

We acknowledge that the peak calling approach was not presented in a way that it would be easy for the reader to understand. (i) We have revised Suppl. Fig. 2a, so it is much clearer. Trivial steps have been removed and the steps of IZ calling are made clear. It is also clearer now that peaks that are close to IZs, which fired earlier in S phase, are excluded when calling IZs, as these peaks may be progressing forks (see bottom part of Suppl. Fig. 2a). We have also updated Supplementary table 1 indicating which sequencing data sets were used in each figure. (ii) Indeed, sample "e" (now named "v") was not used for IZ calling and we have removed it from the figure. (iii) The cutoffs of 50%, 25% and 5% were empirically determined. We note that the cutoff of 25% means that the strongest (top) 25% of bins were used for peak calling (this is now also indicated in the figure). Because the peak to background ratio decreases from early to mid to late replication timing domains, we had to be stricter as we analyzed the later replicating domains and, hence, we focused on the top 25% or top 5% bins for the mid and late S replicating domains, respectively. This is now mentioned in the revised manuscript (lines 557-561). (iv) We now show the same locus in Suppl. Fig. 2b, as in Fig. 1 (chr 8: 50-66 Mb).

c) Once called, the IZ are seemingly classified solely based on the ESC Repli-seq data presented in Sup. Fig. 1c. As Repli-seq largely reflects fork progression rather than initiation, the distinction between mid-S and late-S IZ often relies on which stage had the strongest fork progression in a given genomic region. For instance, replication initiating in the middle of the provided chr7 locus (at app 78Mb) occurs already in the S2 Repli-seq sample. As the authors rightly presents in the discussion and Figure 6, the central challenge is that replication is likely to be initiated from both well-defined origins that fire in most of the cell population and less well-defined 'stochastic origins' that fire in a minor part of the cell population. In the Repli-seq signal fork progression from well-defined origins will dominate, and a 'stochastic' origin giving rise to a mid-S or even early-S IZ with less signal and initiation would be overshadowed by late-S fork progression and therefore be classified as late-S based on the dominating Repli-seq signal. Thus, the current set of samples and IZ identification and classification approach has limited capability to discriminate initiation occurring in mid-S from that occurring in late-S. This is particularly problematic given the emphasis on late-S IZ in the later part of

the manuscript. Perceivably, the solidity of the manuscript would be higher if the authors chose to only discriminate based on direct data (i.e. samples 'l' and 'k') rather than indirect inferences. As acknowledged by the authors, the late-S IZ might as well represent 'stochastic' or low-abundance origins, and this possibility and the procedure limitations should at least be explicit when the origin classification is introduced in the manuscript. If authors decide to proceed with the distinction between mid-S and late-S IZ, they should be introduced as 'putative', 'IZ at mid-S replicated DNA', or similar to more accurately represent the tentative interpretation of these regions.

As the Reviewer states we classify the IZs as early-S, mid-S or late-S IZ according to the replication timing domains where the IZs map; the latter are determined by the Repli-seq profiles, as is standard practice. We agree that the late-S IZs are low efficiency IZs. We did not observe firing of IZs in late-S domains in MSCs or MEFs after release from an aphidicolin block, whereas we observed firing of such IZs after release from an aphidicolin block in mESCs. Thus, there is a difference between firing of IZs that map to late-S domains between mESCs and the other cell types. In the revised manuscript we clarify that the IZs mapping to late replicating domains are less efficient than the IZs that map to the early replicating domains (line 148-150). Further, we clarify that the terms early, mid and late-S IZs refer to the replication timing domain where they map and not to the time they fire in S phase (line 149-156).

2) Comparison between ESC, MSC, and MEF:

A central claim in the manuscript is that ESC are distinct from the other two studied cell lines by having replication initiating very early on in the S-phase at late-S IZ. This claim is solely based on Figure 1g, and this regrettably has several inbuilt assumptions that are not assessed at all. These assumptions need to be addressed as described in the following.

a) First and foremost, the mESC cells underwent additional treatment with HU, which lead to accumulation of signal in certain regions including the late-S IZ. This biases the comparison towards a stronger signal at these weak IZ in the ESC compared to the other cell types. The authors first need to correct this apple-to-banana comparison by presenting data from similar treatments across the cell lines, which in this case also includes accounting for differences in cell cycle progression (see below).

The claim that IZs in late-S replication timing (RT) domains fire in mESCs is based on multiple experiments shown in Fig. 1c, 1f, 1h and Suppl. Fig. 4b. mESCs were synchronized by mitotic shake-off (optionally with CDC7i or ATRi added) or using aphidicolin or thymidine blocks. The same mid-S and late-S origins were identified in all these experiments, also in the absence of HU (Fig.1c(ii)). The MEFs were studied at four different time points after release from an aphidicolin block and the MSCs were studied at two different time points after release from an aphidicolin block. We did not observe firing at late-S IZs in these cells at any time point. Instead, replication of late-S domains appears to involve origin firing throughout these genomic domains, probably in a stochastic manner (Fig. 1h, 1i and Suppl. Fig. 4b).

b) The manuscript omits FACS profiles of the nuclear content to confirm that these cells are indeed G1/S synchronized as the authors claim (line148) - and fundamentally are comparable at all. It is very possible that the cells are in distinct parts of the cell cycle and not comparable at all when used for EdU-seq (more below).

We have now included the FACS profiles in Suppl. Fig. 4a showing excellent G1/S synchronization.

c) The use of a single pulse of aphidicolin to G1/S synchronize ESC, MSC, and MEF in a comparable manner is at best unconventional. Papers from more conventional model systems for cell cycle studies, such as HeLa cells, indicate that aphidicolin leads to arrest both in G1 and at any part of the S-phase that the cell has made it to (e.g. Pedrali-Noy et al, 1980, PMID: 6775308). While the aphidicolin treatment likely leads to a block, it can hardly be called a synchronization and certainly not 'at the G1/S-boundary' (line 148) without a thorough validation of this claim. Authors should use a more appropriate method to synchronize cells at the G1/S boundary, e.g. Palbociclib, and include FACS profiles from these very samples to increase the strength of the data. Ultimately, without a convincing methodology and validation of the underlying assumptions, the authors cannot claim that the difference in the signal at late-S IZ between the cell lines is a biological difference and not an indirect effect of differences in the cell cycle of these cells.

The effect of aphidicolin on cells depends on the concentration used. Low concentrations of aphidicolin slow down DNA replication, but cells are able to complete S phase and then mitosis. The cells will subsequently arrest at the G1/S boundary by a checkpoint response that is activated as cells attempt to enter S phase (see, for example Rossetti et al., 2024; PMID: 39293447; Suppl. Fig. S6A). In the revised manuscript we show by flow cytometry that mESCs, MSCs and MEFs arrest at the G1/S boundary when treated with aphidicolin and after release from the aphidicolin block, all these cells enter promptly into S phase (Suppl. Fig. 4a). Thus, aphidicolin, as used in this study, is a good agent to synchronize mESCs, MSCs and MEFs. In regard to palbociclib, we considered using it to synchronize the cells, but we were not successful, despite trying several times and employing various concentrations of palbociclib (150nM, 500nM, 1µM), as well as other CDK4/CDK6 inhibitors.

d) Given the centrality of this claim, it should be backed by an independent line of evidence, e.g. qPCR-based assays from the regions of interests.

With all respect, we do not consider qPCR as a rigorous assay to monitor origin firing. Most studies on origin firing employ genome-wide methods, such as EdU-seq, rather than qPCR, which can only focus on a small number of regions of interest and where comparisons between different regions is not possible or very difficult to make.

e) The authors only provide EdU-seq data from early time points after aphidicolin synchronization in Figure 1g. Given the differing kinetics of the cell cycle between the cell types used, it is possible that the signal at the late-S IZ would also occur in MSC and MEF if later time points were included. To claim that this is a distinct feature for ESC, then the authors need to rule out that origins within these late-S IZ are firing later in the cell cycle of MSC and MEF.

We examined origin firing in MSCs 30 min and 2 h after release from the aphidicolin block and origin firing in MEFs 1, 2, 4 and 6 h after release from the aphidicolin block. Under similar conditions we observed firing of late-S IZs in mESCs 1.5 h after release from the aphidicolin block (Fig. 1h). When we monitored progression through S phase in the above cells, we can see that progression through S phase takes between 8-9 h (Suppl Fig. 1a and 4a), so the kinetics through S phase are not different in the three cell types.

3) Effects of Oct4 depletion:

a) Oct4 depletion is known to lead to a range of effects, such as cell cycle perturbation and differentiation of ESC, also for the ZHBTc4 model system used (PMID: 19968627; PMID: 14990861; PMID: 25324523). Thus, the reduced EdU-seq signal in Oct4-depleted cells at late-S IZ could be due to an indirect consequence of differentiation. (i) To maintain the claim that Oct4 stimulates origin firing at late-S IZ, the authors need to demonstrate that either no differentiation takes place or that the reported effect occurs in cells that are undifferentiated (e.g. through FACS sorting for Nanog levels). (ii) As it is challenging to get direct evidence of Oct4 binding and inducing initiation at late origins (and likely not a fair demand), authors should more clearly acknowledge the possibility of the relationship being indirect.

We agree with the Reviewer that the induction of differentiation as a consequence of OCT4 depletion is well-known and in the revised manuscript we acknowledge the possibility that OCT4 may act indirectly to modulate firing of late-S IZs (lines 348-349). Of course, this does not affect the conclusion that late-S IZs fire earlier in mESCs than in differentiated cells.

In addition, to minimize the likelihood of indirect effects of OCT4 depletion, the treatment of cells with doxycycline to induce depletion of OCT4 did not exceed 19-25 h (Fig. 5b). In the papers cited by the Reviewer, OCT4 was usually depleted for longer time periods. For example, Lee et al, 2010 (PMID: 19968627) observed a decrease in cell proliferation 48 and 72 h after depleting OCT4, but not earlier. Zhao et al, 2014 (PMID: 25324523) studied a non-transcriptional role of OCT4 on progression of cells through the G2 phase of the cell cycle; thus, this study is not directly relevant to ours. Finally, Hay et al, 2004 (PMID: 14990861) depleted OCT4 using siRNA, which means that cells had decreased expression of OCT4 for more than 24 h. We also bring attention to a study by Xiong et al, 2022 (PMID: 35621159), which shows that depletion of OCT4 in the same system as we used, did not compromise chromatin-bound Nanog levels, 15 h after doxycycline was added to the media (the latest time point examined; Fig. 1A).

b) The evidence provided in Figure 5 is in its current form circumstantial. The authors have to provide actual data on Oct4 from the studied model system, a direct analysis of the Oct4 signal at the late-S IZ, and compare local Oct4 levels in the ZHBTc4 cells (ChIP-seq, ChIP qPCR, or similar). If the presented model is right, then Oct4 levels should be higher in the most affected late-S IZ (red dots in Figure 5e right panel) compared to the less affected late-S IZ. The comparison to public data from a distinct cell line in Figure 5c is not providing evidence suitable for testing their hypothesis.

We understand that the Reviewer is challenging whether the available data provide evidence that OCT4 is physically located at late-S IZs to directly modulate origin firing. We provided publicly available ATAC-seq and OCT4-ChIP data suggesting that OCT4 is located at the late-S IZs (Fig. 5c, 5d), but also acknowledge, as cited in our response to the previous comment, that the effect of OCT4 on origin firing at the late-S IZs may be indirect. We believe that further ATAC-seq and OCT4-ChIP data will not resolve the issue of whether OCT4 acts directly or indirectly to regulate origin firing at late-S IZs.

4) Editing, communication, diligence, and figures:

The manuscript editing seems premature in several ways and needs adaptations to cater for a broad audience.

a) Rationales for using different inhibitor regimes are in some cases presented much after the initial use of the inhibitors.

This is particularly clear for CDC7-inhibition, which is presented in Figure 1, but the rationale only follows in the results section describing Figure 3. Presumably, authors did a major restructuring of the manuscript late in the process without ensuring that information is presented in a logical order. Also, CDC7 inhibition is listed as CDC7i, TAK-931, and TAK in figures and the result section, and authors need to clarify if these are the same treatments and if so, make the text consistent. Finally, methods are lacking when it comes to description of the inhibitor treatments, and there is allegedly no description of the concentrations used when inhibiting CDC7. Relatedly, concentrations of e.g. HU are also inconsistently listed or left out in the manuscript and figures.

We thank the Reviewer for pointing out these omissions. We have revised the manuscript, as suggested by the Reviewer. Concentrations of inhibitors are listed in the Method sections (lines 396-400) and in the Figure Legends. Abbreviations are also cited in the Figure Legends.

b) In figure 1 and Supplementary Figures 1 and 2 the authors present a complex set of combined treatments labelled 'a' to 'n'. While it is praiseworthy to use the small overview timelines (generally in many of the figures) as well as a 'unique identifier' for the many conditions, the latter is not forthcoming for the readers' ability to digest the panels and findings. The current version of Figure 1 and supplement almost feels like a cryptography exercise where dozens of cross-references need to be made. The authors should help the readers further by using more informative labels that also describe the characteristics of the samples. This can certainly be combined with the use of a unique identifier, but it would enhance clarity to use one that is distinct from the lettering used to label the panels.

We understand the point of the Reviewer, but using names to refer to the samples becomes complicated given the various treatments that the samples were subjected to. For example, MSO-Harv7h-CDC7i3-6-HU6.5-7, could be used to refer to cells synchronized by mitotic shake-off, harvested at 7 h and treated with a CDC7 inhibitor between 3 and 6 h and with HU between 6.5 and 7 h.

c) The description of the data visualization done in Figures 3-5 in methods is absent, and the bash script for peak calling is not accessible in the provided link (link broken, 404 error).

We apologize for this omission. We have added the description for the EU-seq in the Methods section (lines 635-654) and the link has been fixed (line 659). The Edu-seq average signal was already described (now in lines 590-595).

d) In Figure 1 and the corresponding supplement, the authors present the foundation for the central claims made in the manuscript. The work however only presents EdU-seq profiles from a single locus at chr8, and much of the conclusions are based on enrichment profiles a two alleged late-S IZ. Key conclusions in these figures should be substantiated by generalized visualizations like the heatmaps, average signal, and scatter plots presented in Figures 2-5.

We have now added heatmaps, average signals and/or scatter plots, as relevant, for all the main experiments of the manuscript. These are shown in Figs 1g, 1i, S3a, S3b and S4b.

OTHER POINTS

1) The title of the manuscript is misleading. Since the manuscript solely focuses on one of the pluripotency factors, Oct4, the title should reflect this. Generalization to 'pluripotency factors' is a stretch.

We accept the proposition and changed the title of the manuscript (line 5), as suggested by the Reviewer.

2) In the abstract, the authors claim that the spatiotemporal program of replication is not well-defined (line 29). This seems inappropriate given the solid amount of work on this, and it is also contradicted by the authors' own statements later in the manuscript, e.g. in lines 56-59!

Indeed, we have edited this part of the abstract (line 29).

3) The knowledge gap in the introduction could be described more clearly and thoroughly.

This is now better described (lines 80-85).

4) There are oddities in the EdU-seq signal profile of several of the samples. The authors should acknowledge this in the manuscript and explain those. This includes the enrichment in G1 samples in Sup Fig 1c, elevated signal 2 h after synchronization at Late IZs in Sup Fig 2c, abundant signal at late replicating regions in Figure 1c sample 'a' (e.g. in the middle of the chr8 locus), abundant signal in late replicating regions in the MSC 0.5h sample in Figure 1g (e.g. at the start of the locus).

The enrichment in G1 fraction is due to the high sensitivity of the EdU-seq method and the fact that the fraction includes the very first moment of S-phase, as seen in the schemes in Suppl. Fig. 1b. We were also surprised by the presence of replicating activity in late regions in the early moment of S-phase. Contrasting this with an article from the group of Massimo Lopes (Ahuja et al 2016), we could hypothesize that because ESCs do not manage to replicate completely the whole genome in a given cell cycle, some late regions remain unreplicated. The activity in late regions could then correspond to regions that did not complete replication in the previous cell cycle, and which resumed in the current S-phase. Although this is a very intriguing finding, the response to this conundrum goes beyond the current scope of the manuscript.

5) EdU-seq and EU-seq normalization seems inconsistently mentioned. In some figures and in the IZ calling section of the methods 'BPM' is mentioned, but in other figures these labels are absent. There is a mentioning of log₂+1 transformation (line 531, but it is unclear which plots where the signal is log-transformed, and it is preferable to have such information in the relevant figure(s).

This has now been corrected, as the Reviewer suggested.

6) The overall quality of Figure 4 is low. This needs to be addressed. Specifically, Figure 4a culture dish, 4c and 4d heatmaps needs to be updated with higher quality versions.

These panels have now been updated.

7) The clustering done in Figure 4c is unsuited to conclude that “Of the early-replicating genes, more were transcriptionally active in mESC than in MEFs”. Dedicated analyses of this should be made to support this claim, and a quantitative approach would convey this better. For example, a box or violin plot visualization of nascent RNA levels of each RT domains would be nice.

Following the reviewer’s advice, we have now added a figure showing the full comparison of nascent transcription between mESCs and MEFs (Suppl. Fig. 7c).

8) Stars (*) in Sup Fig 2b are not explained.

We apologize for this. They referred to an earlier version of the figure and have now been removed.

Reviewer #1 (Remarks on code availability):

The code link is broken, so a review could not be done.

The full code is now available.

Reviewer #2 (Remarks to the Author):

Reviewer #3 (Remarks to the Author):

In this manuscript by Rodriguez-Carballo and colleagues, the authors show that mESCs fire mid to late initiation domains early when compared to MEFs or mesenchymal stem cells. Interestingly, this change of replication timing is mediated by pluripotency factors like OCT4 and correlates with chromatin accessibility. This is an interesting observation but the mechanisms by which the pluripotency factors are driving this change of replication specifically in mESCs are not elucidated. Below are the main comments which should clarify some of the outstanding questions:

1. What percentage of mid-late IZs are firing in ESCs? the authors should have a cumulative plot showing it in the main figure.

We have now added cumulative plots of the signal distribution for all tracks shown in Fig1; see new Fig.1g, I and Supplementary Fig. 3 and 4.

2. Fig1F, after CDC7i treatments, the overall origin firing seems to be diminished (e vs j). Could the reduction of mid and late IZ firing just be due to that?

The two samples: e (now labeled v) and j (now labeled x) in Fig. 1f are plotted at the same scale (0-3). The cells treated with the CDC7i (sample j/x) have better resolved peaks corresponding to progressing forks, because origin firing was restricted to 0-3 h after mitotic shake-off (since the CDC7i was added 3 h after mitotic shake-off). In contrast, in the sample without CDC7i treatment (sample e/v), some cells fired origins within the first 3 h, but in other cells, these origins fired later, resulting in a stronger overall signal and less well-defined peaks.

3. Can the firing of mid and late IZs be a result of synchronization methods which induce replication stress for example use of high HU doses which completely stalls the forks which in turn activates origins locally in mESCs specifically? Can the authors dissect that out?

We do not think that this is the case, because we saw origin firing at the mid-S and late-S IZs even in cells not treated with HU (for example, Fig. 1c, sample b/ii) and also in cells synchronized by thymidine or aphidicolin (Fig. 1f, samples m/xiii and n/xiv).

4. The authors show in Fig 4 that the mid and late IZs are marked by open chromatin (not necessarily co-relating with transcription). However, this chromatin accessibility at mid and late IZs seems to be dependent on OCT4. How do the authors envisage OCT4 mediating chromatin accessibility? This mechanism needs to be dissected out as pluripotency factors driving changes of replication timing of mid and late IZs is the most interesting part of the entire manuscript.

We propose that the genomic location of replication origin firing greatly depends on chromatin accessibility. Pioneer factors, such as OCT4, have been previously shown to displace nucleosomes and to allow other transcription factors to bind to these regions keeping them “open” (Soufi et al 2012; 2015; Chen et al 2016; Echigoya et al 2020). This window of opportunity can then be seized by replication factors (ORC, etc) for origin licensing and firing. We think that it is beyond the scope of this manuscript to explore how OCT4 opens chromatin.

Reviewer #4 (Remarks to the Author):

Reviewer #5 (Remarks to the Author):

In Rodriguez-Carballo et al., the authors thoroughly characterize DNA replication initiation throughout early-, middle- and late-S-phase and explore its relationships with experimental perturbations and broader chromatin organization.

The authors specifically focused on mouse embryonic stem cells (mESCs) for most of their experiments as the short G1-phase of mESCs allowed for the authors to perform cell cycle sorting by capturing cells in 2hr intervals after mitotic exit. This sorting procedure critically enabled the authors to measure initiation, and ensuing replication fork progression, with EdU-seq protocols which provide higher resolution than competing Repli-seq and OK-seq assays. After characterizing initiation zones (IZs) mESCs, mMSCs, and MEFs, the authors began to probe specific relationships between replication timing (RT) and IZs through perturbations such as CDC7 inhibition (via TAK-931), dNTP depletion, ATR inhibition, and CDK1 inhibition. The authors comprehensively rationalize why each experiment was performed and relate their results to broader literature & prevailing theories on the role of different checkpoints (i.e. Fig 3c).

In the second half of the paper, the authors transitioned to examining how replication varies between epigenetic and transcriptional states. The authors make valuable use of external ATAC-seq and other epigenetic data to support their claim that open chromatin features are present at IZs. Finally, the paper concludes with an investigation into how pluripotency factor binding is enriched across all IZs and that depletion of OCT4, through a dox-inducible system, led to decreased chromatin accessibility and origin firing efficiency at late IZs.

Overall, the data and biological insights from this paper represent a meaningful contribution to the field which will spur future investigation into how late-firing origins differ from early firing ones and their relationships with epigenetic modifiers such as OCT4 and other pluripotency factors. In spite of my general enthusiasm, there were several areas in the paper which could be strengthened through more rigorous statistical analysis. I would particularly like these comments to be addressed before recommending publication.

Major comments

1. L127-152: Results from Fig 1c, f, g are mostly limited to arrows pointing at very small peaks within a ~15Mb region of chr8. These results would be significantly strengthened, and easier to interpret, if the authors added an additional panel(s) to Fig 1 which demonstrates that these trends are conserved across the entire genome (perhaps something like Figs 2b, 2g, 5e?). Looking genome-wide would also enable the authors to strengthen their claims by testing for statistical significance, effect sizes, etc.

We have addressed this comment by adding genome-wide analyses of EdU-seq data, as suggested by the Reviewer (see Fig. 1g, 1i and Suppl. Figs. 3 and 4).

2. L217-224: The clustering approach used by the authors is not statistically rigorous enough to claim that certain RT domains are enriched for certain transcription levels (L219-221) or that early-replicating genes were more transcriptionally active in mESCs than in MEFs (L221-223). The clustering approach might be okay for visualization purposes in Fig 4c but this is another result that would be strengthened by statistical hypothesis testing and an additional figure panel that more clearly supports the claim on L221-223

We have performed statistical analysis of our EU-seq data (lines 236-249, 635-654) and added a new Suppl. Fig. 7 complementary to the main figure. In MEFs, transcription is restricted to early and mid-S replicating domains, while, in mESCs, transcription is also detected at late-S regions.

3. L236-237: Another example where statistical testing would better show that there is still enrichment of epigenetic marks AND that effect size of this enrichment is lower in mid/late IZs compared to early IZs.

We have performed the requested statistical analysis, which shows significant enrichment (Supplementary Fig. 8b).

4. L258-269: I like the fact that the authors performed genome-wide analysis via scatterplot in Figs 5e and S6d,e; however, I would prefer if the authors reported p-values to compare “observed vs expected” rather than the $y=mx+b$ linear regression coefficients and the angle between linear regression fit of the data vs expected. One suggestion would be to compute the residual values between the observed data points and expected values → fit a linear regression model to the residuals → report a p-value for a hypothesis test whose null hypothesis is that the slope is zero. If using python, this can be easily done using the p-values reported by scipy’s linear regression tool <https://docs.scipy.org/doc/scipy/reference/generated/scipy.stats.linregress.html>

We thank the Reviewer for this comment. As suggested, we have now implemented a residual regression analysis (lines 613-633), as described in the Materials and Methods section under the subsection Residual regression analysis. This approach involves computing the residuals between OCT4-OFF and OCT4-ON values, fitting a linear regression model to the residuals, and testing whether the slope significantly deviates from zero. The corresponding analysis was performed using R, and the script is available in the GitHub repository associated with this manuscript. The new panels associated to this analysis (depletion of OCT4 for 12h and 18h) are shown in the new Fig. 5e. We have kept the previous analysis as well (see Suppl. Fig. 9d-f).

5. Change “pluripotency factors” to “OCT4” in the title or replace/remove the “pluripotency factors enhance the” clause all together. While the authors revealed that all four pluripotency factors (OCT4, SOX2, KLF4, NANOG) are enriched at early, mid and late IZs in Fig 4h, only OCT4 was further investigated in Fig 5 to show that its absence delayed the firing of mid- and late IZs.

We have changed the title (line 5) and agree that it is better.

Minor comments

1. L65-73: Consider citing Emerson et al., Nature, 2022 somewhere in this introduction paragraph as they seem to have investigated late IZs in human cells, albeit at 50kb resolution since they used RepliSeq.

Emerson et al 2022 is now cited in the introduction (line 61).

2. L697-702: Define the x-axis for Fig S1a. I think it’s ploidy but should be specified in the legend and axis label.

Thank you, the term ploidy has now been added to the x-axis.

3. L105: Specify the resolution of the RT profiles you obtain here. You mention that they’re 10kb in the methods (L490)

but this seems to be an important point as you mention the high-resolution of this data as a key differentiating factor from previous Repli-seq studies of late IZs. (lines 65-73).

The resolution is mentioned in the text (lines 112).

4. L108: “Repli-seq experiments extrapolate time from genomic content” is a confusing phrase. Please reword.

This concept is now explained to justify an approach on which we could control time progression since S-phase entry (lines 116-118).

5. L108-119: Spit the heatmap in Fig S1e into a new panel S1f (keeping the two scheme diagrams in S1e) so that you can separately reference experimental design vs heatmap results in the main text. Also, only show the upper left or lower right triangle of this heatmap since it is symmetric. Specify the type of correlation used in color bar or legend.

Because the original heatmap may allow for an easier comparison between the Repli-seq and EdU-seq experiments, we decided to keep it and thank the Reviewer for her/his understanding.

6. L130-152: Instead of assigning letters for each track in Figs 1b-f, assign roman numerals or some alternative scheme. Having letters for both tracks and figure panels can be confusing when reading the main text.

We changed the numbering of the tracks, as suggested (lines 142-167).

7. L249-254: Reference Fig 5b somewhere near here as mention of Fig 5b seems to be missing from the main text.

Fig 5b is now referenced (line 284); thank you for noticing this.

8. L550-553: More code used to analyze data and produce figures should be published on github. The authors released code for peak calling; however, code to produce Fig 2b,c,g, 4c,d, 5e, etc. should also be released.

Thank you for your comment. We would like to clarify that no custom code was used to analyze the data presented in Figures 2b, 2c, 2g, and 4c, 4d.

Figure 2b shows the average EdU signal of early IZ fork progression, with or without HU treatment, across various timepoints. These signals were obtained using the ComputeMatrix function of deepTools (<https://deeptools.readthedocs.io/en/latest/content/tools/computeMatrix.html>), analyzing regions spanning 1 Mb upstream and downstream of each fork at 1 kb resolution. Fork speed was estimated by measuring the distance that the average signal peak (summit of the curve) traveled over time. Specifically, this corresponds to the distance in kilobases divided by 120 minutes (i.e., the 2-hour timepoint).

Figure 2g was generated using an analogous approach to that of Figure 2b.

Figure 4c presents a heatmap visualization based on EU-seq analysis, specifically the Transcript Quantification and Stratification method described in the *Materials and Methods* section (lines 635-645). This analysis relied entirely on the publicly available tool TPMCalculator for transcript quantification—no custom scripts were used.

For Figures 2b, 2c, and 2g, publicly available Lamina-associated domain (LAD) data were transformed into a binary format (0 = no LAD, 1 = LAD) and visualized using deepTools.

Regarding the linear correlation analysis of EdU-seq and ATAC-seq data in the mESC inducible OCT4 system: this has now been updated to a residual regression analysis (lines 613-633). The in-house R code used for this updated analysis is now publicly available.

We hope this clarifies the methodology and data analysis procedures used in these figures.

Reviewer #5 (Remarks on Code Availability):

Only one script for peak calling is provided. The authors could also include other code useful for reproducing the results, for instance the EdU-seq or fork progression code.

These codes are now provided.

REVIEWER COMMENTS

Reviewer #1 (Remarks to the Author):

In this revised manuscript, the authors have addressed some of our previously presented points while leaving central points unaddressed or only partially addressed.

It is very possible that mESC differ from MEFs and MSCs in their replication timing regulation and that Oct4 accelerates local replication as the authors claim. This would not be a surprising finding given the existing knowledge in the field.

However, without the right evidence it is not possible to discriminate between this and other models, and we find that the overall methodology does not provide the required rigor.

This undermines the strong conclusions made in the paper, and in its current state several of the main points presented in the abstract are not substantiated. At the same time the authors tend to drive a preferred interpretation and not entertain alternative explanations. We find that the combination of the one-sided reporting and the extent of the methodological weakness to be material, and the revised manuscript provides little new evidence to reduce our concerns.

We do understand and acknowledge that what the authors attempt to do is not technically trivial and that there are limitations on what the different cell types allow in terms of synchronization and assaying.

Given these limitations and inability/unwillingness to provide substantiating evidence it would be appropriate to present data with a more open reporting style that **acknowledges limitations and alternative explanations more explicitly**.

A bare minimum of diligence would require authors to tone down claims in concordance with the points presented below and to provide a transparent reporting on these confounding factors – **both when results are presented and discussed and in a separate Study Limitations section**.

Main unresolved points:

1. The authors **overconclude and oversell in the abstract**. This either demonstrates a disqualifying lack of care or a drive to push their claims beyond what the data supports. Main claims in the abstract were not and are still not supported by data, and in the absence of supporting data, the **abstract needs to be modified** to reflect actual findings. E.g.:

a. The statement that “we identified DNA replication initiation zones (IZs) in mESCs, mesenchymal stem cells and mouse embryo fibroblasts (MEFs)” is factually wrong. **The authors only identify IZ from mESCs** and then study EdU-signal levels at these sites in the other cell types. It is concerning that such simple matters are not congruently presented.

b. Similarly, as the authors acknowledge in their point-by-point response then they **only investigate Oct4, yet** generalize to pluripotency factors in the abstract: “This change in the replication timing program, which did not lead to shortening of the length of S phase, was driven by pluripotency transcription factors, notably OCT4...”.

c. The statement that “**Uniquely in mESCs, we observed, within 1-2 hours of entry into S phase**, origin firing at IZs that mapped to the mid and late-replicating genomic domains” is based on apple-to-banana comparisons. This was pointed out in our previous set of comments, and the problem is aggravated when the authors end up using the resultant absence of evidence to conclude evidence of absence. There are central dissimilarities in both data and methods: Firstly, the assays used to observe initiation in the cell types differ on central points and only mESC were treated with HU, whereas none of the tracks from the MEF and MSC had such a treatment. Secondly, IZs were only ever identified in mESC, and the replication in MEFs and MSCs was monitored as these sites. This results in a lopsided comparison where selection biases may affect conclusiveness. To conclude that this is a unique trait for mESC authors need to identify IZs in MEFs and MSCs using an identical approach and observe a lack at these sites. Without data that allows a more direct comparison, **the statement should be softened or left out**.

d. The use of ‘**change**’ in the statement “**This change in the replication timing program**” implies a that a comparison is performed across two related states and not across three distinct cell types. Moreover, given the limitations described above, this difference might as well be due to methodological differences.

Based on the remarks of the Reviewer, we have re-written the abstract. All the statements mentioned above have been addressed according to the guidance of the Reviewer.

2. In their response, the authors claim that the effect of Oct4 on cell cycle in ES cells is negligible as they assay cells 19-25 h after induction, while proliferation decrease occurs at 48h and 72h in the previous work with the title “Oct-4 controls cell-cycle progression of embryonic stem cells” (PMID: 19968627). However, at 48 h the published work reports less than one third the number of control cells (PMID: 19968627, Fig 1B). It is inconceivable that the massive preceding reduction in proliferation occurred in the final hour of the assay or that it was not accompanied by perturbations of the cell cycle long before this time point.

The authors should with relative ease be able to demonstrate whether this is the explanation for their observations but chose not to provide such **data for the revised manuscript. This should be provided.**

In their article, Lee et al (2010, PMID: 19968627) show in Fig. 1B a clear decrease in ES cell numbers at 48 h after adding doxycycline to deplete Oct4 expression. However, at 24 h there was no decrease in ES cell numbers (also shown in Fig. 1B). In another article, Zhao et al (2014, PMID: 25324523) showed that in ES cells OCT4 has a non-transcriptional role to delay entry into mitosis and that reducing OCT4 levels leads to premature mitotic entry and chromosomal abnormalities. Together these results suggest that the first cell cycle, up to the G2 phase, is not affected by OCT4 depletion. In our manuscript we state that loss of OCT4 does not affect “the kinetics of entry into S phase after synchronization by MSO” (lines 284-286).

The purpose of our manuscript is not to analyze all roles of OCT4 during the cell cycle, but to propose that the OCT4 pioneer action may stimulate initiation of DNA replication at specific IZs. We believe that dissecting the role of OCT4 in cell cycle progression, DNA damage control and apoptosis, is beyond the scope of the current manuscript.

3. Readers may get the impression that e.g. the ‘Late IZ’ are firing late. The authors now do clarify that this is not their claim. And it would be appropriate to make this clearer throughout the manuscript, e.g. by also clarifying this early in the reporting. The recurrent use of e.g. ‘late IZ’ throughout the manuscript e.g. lines 230, 275, 326, 327, 346 without this context gives the impression that the authors believe that initiation occurred at the indicated stage. The authors should throughout the manuscript make sure that initiation timing and initiation at loci predominantly replicating at a given stage is not conflated. Although wordier, one way would be to use the term ‘IZ in late replicating regions’ or ‘predominantly late replicating IZ’.

We thank the Reviewer for acknowledging that we now clarify that the term “Late IZ” is defined as referring to the replication timing (RT) domain to which the IZ maps. We thank also the Reviewer for proposing terminology that will be the least confusing for the reader. After careful reflection, we consider that the terms “late RT IZ”, etc, convey the correct meaning and are also short, making the text easier to read. In the previous version of the manuscript, there were instances in the text, which required fixing of these terms (see lines 185, 265, 276, 304, 305, 346, 367, 374).

4. We understand the response provided regarding the IZ calling and agree at the data provided in the response figure reduces the concerns at the illustrated locus. However, the challenge of using algorithms and pipelines to search the entire genome, is that even in the best setup noisy data and edge cases tend to lead to false positives and negatives. This possibility is not handled well in the current version of the manuscript, and the authors have no figures or analyses that on a genome-wide scale address the robustness of the called IZs. Neither do they seek to use independent methodology (such as qPCR, which they describe as unsuitable in the response). Thus, conclusions end up relying on one line of evidence without any potential validation.

a. The authors should discuss (e.g. in discussion or **study limitations**) **how both false positive and negative calls will affect conclusions.**

We have added a section to the Discussion addressing the potential presence or absence of false-positive and false-negative calls (see lines 335–344). In this section, we clarify that, given our stringent exclusion criteria, we are more concerned about false negatives (i.e., not capturing all possible efficient IZs) than about false positives.

b. Moreover, they should seek to **validate the three categories of IZs, e.g. using the provided OK-seq data** – conceivably IZs would have a step change in OK-seq directionality. This can and should be analyzed on a genomic level, and it would be helpful for the reader to know that e.g. 70% of the IZ in late replicating regions were associated with a change in OK-seq directionality.

We have now added a new panel (Supplementary figure 3c), which shows heatmaps and average plots of OK-seq directionality along the IZ datasets. These graphic approach shows a clear correlation between our called IZs and OK-seq data.

c. Finally, the 500 kbp zone used for exclusion seems tight. In the provided examples this seems ok although the shoulders of the progressing fork signal in sample (xi) seems to be within the 500 kbp zone. For more jagged/noisy signal, it is possible that peaks would be called from these flanks just outside the zone. This might also not be the case, but the reader needs figures that allows her/him to judge this themselves, and currently it is hard to make a general assessment. The authors should therefore provide supplementary figures that illustrate and substantiates that this is an appropriate exclusion zone. They should also plot the density of late S IZ as a distance of mid S / early S IZ (and the same for mid S IZ relative to early S IZ). If the exclusion zone is appropriate, there should expectedly a rather flat distribution of IZs outside the 500 kbp zone.

We performed the analysis suggested by the referee. We have now added a boxplot and a summary table in Supplementary Fig. 2C. For each IZ, we calculated the distance to the nearest IZ belonging to a different RT domain. In the boxplot, we can see that very few IZs are detected near the 500 kb threshold we established as the exclusion zone. Indeed, the majority of the identified IZs are located more than 2 Mb away from an IZ in a different RT domain.

5. mESC are heterogenous and widely known to spontaneously differentiate, and this is very pronounced when grown in serum (as done in this manuscript), while newer culture conditions relying on 2i medium reduces this. Therefore, the observed differences between mESC and MEFs / MSCs could be due to a more heterogenous cell population. With the used IZ classification, IZs end up being determined based on the RT profile in the dominant cell population. The manuscript should more directly acknowledge this, e.g near the discussion in lines 300-303.

The reviewer is correct in highlighting the heterogeneity of mESC cultures. In fact, we think that the signal of the late RT IZ would be higher, if all the cells in our cultures were pluripotent. This would fit with the absence of EdU-seq signal at these genomic positions in MEFs and MSCs, i.e. in non-pluripotent cells. Of course, we have no data to support this hypothesis. Given this, anything we write in the Discussion on this topic would be entirely speculative.

Minor points:

6. In the response the authors write: “we could hypothesize that because ESCs do not manage to replicate completely the whole genome in a given cell cycle, some late regions remain unreplicated. “ as a possible explanation for some of the aberrant profiles. This would be a huge controversial finding on its own, and it is plausible that other more mundane explanations exist, e.g. impurities in the assayed fraction. The aberrant tracks should be mentioned in the manuscript along with possible explanations.

During the data analysis phase, we discussed the origin of this signal but were unable to reach a definitive conclusion. We could not identify an experimental approach to readily distinguish between "replication resuming in unreplicated regions", as suggested previously by Ahuja et al. (2016; ref. 41 of our manuscript), versus carry-over of late S phase cells or of mitotic cells undergoing mitotic DNA synthesis (MiDAS), which we think is the most likely explanation. We mention both possibilities in the Results section (lines 125–129).

7. We did not ask for exhaustive naming or complex names such as ‘MSO-Harv7h-CDC7i3-6-HU6.5-7’, just labeling. If the authors disagree on the need for memorable and understandable labels, then at least restructure the layout of figure 1, so that the scheme in e is on the left side of the tracks in f (and similar for b, c). This structure is already used in other figures.

We tried several approaches to fit all schemes and tracks in Figure 1, and our proposed setup is the only one that allows us to display a genomic region large enough to include different types of IZs. Placing the schemes on the left would force us to split the large genomic area into smaller regions, losing the comprehensive view of early, mid, and late RT IZs, along with their associated firing dynamics.

8. Suppl. Figure 2a contains BMP instead of BPM a couple of occasions.

Thank you for noticing this typo. We have now corrected it.

Reviewer #2 (Remarks to the Author):

Reviewer #3 (Remarks to the Author):

The authors have addressed all my concerns sufficiently and I can recommend the manuscript for publication.

Reviewer #4 (Remarks to the Author):

Reviewer #4 (Remarks on code availability):

The authors used code / bioinformatics tools from other studies to produce the most of the figures from their manuscript. The original code from this manuscript pertains to peak calling which has been deposited into a github repository. This repository could be improved through the addition of a README.md file which specifies the R package dependencies and instructions for running their scripts.

A ReadMe file has now been included in the github repository.

Reviewer #5 (Remarks to the Author):

The authors have addressed my concerns.

Reviewer #5 (Remarks on code availability):

The provided code only includes a peak calling script. It would be more optimal to also include a .Rmd or other code file that shows how the authors created the figures.

A ReadMe file has now been included in the github repository.

REVIEWER COMMENTS

Reviewer #1 (Remarks to the Author):

The authors have generally performed revisions that address previously presented points.

This includes a rewritten and substantially changed abstract, where the major claims now better reflect the conclusiveness of the data.

In addition, the authors have included several passages that present and consider alternative interpretations.

In general, I can support publication.

A few minor points remain, but these can be addressed without further reviewing:

We thank the Reviewer for these positive comments.

The title of the final section of results (line 275) still states “Pluripotency factors act as pioneer factors for mid and late IZs”, as they do not investigate role of any other pluripotency factor other than OCT4 in this section, it should be updated to ‘The pluripotency factor Oct4’ or similar.

This has been addressed, as the Reviewer proposed (line 271).

Given the centrality of Oct4 for the claims made in the manuscript, it would be appropriate to include a deeper discussion of the possibility of Oct4 having other effects on cell cycle progression including the points made in their response to the previous major point 2.

We now discuss the effects of Oct4 on the cell cycle in the Discussion (lines 366-374).